# Permeant fluorescent probes visualize the activation of SARM1 and uncover an anti-neurodegenerative drug candidate

Wan Hua Li[1,2], Ke Huang[3], Yang Cai[4], Qian Wen Wang[1], Wen Jie Zhu[1], Yun Nan Hou[1], Sujing Wang[1], Sheng Cao[5], Zhi Ying Zhao[1], Xu Jie Xie[1], Yang Du[5], Chi-Sing Lee[3]\*, Hon Cheung Lee[1]\*, Hongmin Zhang[4]\*, Yong Juan Zhao[1,2,6]\*

[1]State Key Laboratory of Chemical Oncogenomics, Key Laboratory of Chemical Genomics, Peking University Shenzhen Graduate School, Shenzhen, China; [2]Ciechanover Institute of Precision and Regenerative Medicine, School of Life and Health Sciences, The Chinese University of Hong Kong, Shenzhen, China; [3]Department of Chemistry, Hong Kong Baptist University, Kowloon Tong, Hong Kong, China; [4]Department of Biology, Southern University of Science and Technology, Shenzhen, China; [5]Kobilka Institute of Innovative Drug Discovery, School of Life and Health Sciences, The Chinese University of Hong Kong, Shenzhen, China; [6]Shenzhen-Hong Kong Institute of Brain Science-Shenzhen Fundamental Research Institutions, Shenzhen, China

*For correspondence:
cslee-chem@hkbu.edu.hk (C-SL);
leehoncheung@gmail.com (HCL);
zhanghm@sustech.edu.cn (HZ);
zhaoyongjuan@cuhk.edu.cn (YJZ)

**Abstract** SARM1 regulates axonal degeneration through its NAD-metabolizing activity and is a drug target for neurodegenerative disorders. We designed and synthesized fluorescent conjugates of styryl derivative with pyridine to serve as substrates of SARM1, which exhibited large red shifts after conversion. With the conjugates, SARM1 activation was visualized in live cells following elevation of endogenous NMN or treatment with a cell-permeant NMN-analog. In neurons, imaging documented mouse SARM1 activation preceded vincristine-induced axonal degeneration by hours. Library screening identified a derivative of nisoldipine (NSDP) as a covalent inhibitor of SARM1 that reacted with the cysteines, especially Cys311 in its ARM domain and blocked its NMN-activation, protecting axons from degeneration. The Cryo-EM structure showed that SARM1 was locked into an inactive conformation by the inhibitor, uncovering a potential neuroprotective mechanism of dihydropyridines.

## Introduction

Axon degeneration (AxD) occurs in most neurodegenerative disorders (*Coleman and Höke, 2020*). Sterile Alpha and TIR Motif–containing 1 (SARM1) acts as a main effector in this process (*Osterloh et al., 2012*) and its depletion significantly attenuates AxD (*Geisler et al., 2016*; *Osterloh et al., 2012*; *Turkiew et al., 2017*). SARM1 controls AxD through its enzymatic activity (*Essuman et al., 2017*). It is self-inhibitory and is activated by nicotinamide mononucleotide (NMN) (*Zhao et al., 2019*), resulting in depletion of the intracellular NAD-pool (*Essuman et al., 2017*; *Zhao et al., 2019*). However, recent studies suggest that nicotinamide adenine dinucleotide (NAD) itself is an inhibitor of SARM1 activation and the balance between NMN and NAD controls the activation of SARM1 (*Figley et al., 2021*; *Jiang et al., 2020*; *Sporny et al., 2020*).

It should be noted that SARM1 is not just a simple NADase activated to deplete the cellular NAD. We have documented that SARM1 is a multifunctional enzyme with properties similar to CD38, a universal signaling enzyme possessing not only NADase activity but also catalyzing both the

cyclization of NAD to cyclic ADP-ribose (cADPR) and the exchange of nicotinamide in NADP with nicotinic acid to produce nicotinic acid adenine dinucleotide phosphate (NAADP) (*Zhao et al., 2019*). Both cADPR and NAADP are messengers regulating calcium mobilization in the endoplasmic reticulum and the endo-lysosomes, respectively (reviewed in *Galione, 1994*; *Lee, 2012*; *Lee and Zhao, 2019*). The catalytic similarities and its ubiquitous presence in non-neuronal cells suggest that SARM1 may be a calcium signaling enzyme as well.

Since SARM1 is important in axon degeneration and potentially other physiological processes as well, we thus aim to design and synthesize probes for visualizing SARM1 activation in live cells and to screen drug library for potent inhibitors.

## Results

### Probe design, synthesis, and characterization

We focused on its base-exchange reaction for designing specific probes for SARM1 and had shown that pyridyl derivatives can readily serve as substrates (*Graeff et al., 2006*; *Lee and Aarhus, 1997*). We thus conjugated various styryl derivatives to pyridine to produce a series of conjugates (PCs) as fluorescent probes for SARM1 activity (*Figure 1A*). We reasoned that conjugating the electron-rich styryl derivative with pyridine should provide a donor-π-acceptor framework (*Pawlicki et al., 2009*; *Figure 1B*). The positive charge of the pyridinium moiety of the product should delocalize over the conjugated π-system and lead to fluorescence changes (*Figure 1A*). Pyridine conjugates (PC1-9, *Figure 1—figure supplement 1*, *Figure 1—figure supplement 11A*) were synthesized using the Pd-catalyzed cross-coupling strategy with yields ranging from 33.5 to 85.0%. The synthesis details are in the 'Materials and methods' section and product characterizations are shown in *Figure 1—figure supplements 2–10*.

The PC probes were tested using a recombinant SARM1 with the N-terminal mitochondrial-localizing peptide truncated, SARM1-dN (*Zhao et al., 2019*) (described in *Figure 1—figure supplement 11B*), with NAD as the acceptor of base-exchange and NMN as an activator. As shown in *Figure 1—figure supplement 12*, significant shifts in UV-vis spectra were observed in the reactions with the oxygenated derivatives (PC5-8, *O*-series), but not the nitrogenated derivatives (PC1–4, *N*-series), nor PC9 with diene as linker. The emission spectra of the reactive *O*-series showed steady increase as the reaction progressed (*Figure 1—figure supplement 13*, spectra; *Figure 1D*, kinetics; *Figure 1—figure supplement 11C*, initial rate), with PC6, the chemical structure shown in *Figure 1C*, exhibiting the largest fluorescence increase (*Figure 1D*).

The time course study for the reaction involving PC6 showed that the UV absorption decreases at 330 nm but increases at 400 nm with an isosbestic point at 350 nm (*Figure 1—figure supplement 12*; *Figure 1E*). Corresponding to the absorbance change was the red shift in the fluorescence spectra, from the emission maximum at 430 nm of PC6 to 520 nm of PAD6 (*Figure 1E*).

The conversion of PC6 to the exchange product, PAD6, was verified by purifying it using HPLC and characterized by high resolution mass spectrometry (HRMS) (*Figure 1F*). The remarkably large spectral changes are anticipated from our design, as the pyridine ring becomes positively charged after its exchange into NAD (*Figure 1F*, inlet), a much stronger electron acceptor in the D−π−A structure, thereby increasing intramolecular charge transfer and shifting the emission maximum by over 100 nm. The conversion-induced spectral changes were consistent with the spectra of the HPLC-purified products, PAD6 (*Figure 1G*).

The observed spectral changes showed a linear dependence on NMN, with as low as 10 μM being effective (*Figure 1H*), confirming that SARM1 is an auto-inhibitory enzyme activated by NMN (*Zhao et al., 2019*). The fluorescence increase was also proportional to the amount of NMN-activated SARM1 (*Figure 1I*), with a detection limit of 48 ng/mL. As an in vitro assay for SARM1, PC6 provides more than 100-fold higher sensitivity over other commonly used probes, such as εNAD, NGD, or NHD (*Figure 1J*).

In addition to sensitivity, PC6 also shows exquisite selectivity toward SARM1 versus CD38 and *N. crassa* NADase (*Graeff et al., 1994*). All three possess NADase activity as detected by εNAD (*Figure 1K*), but only SARM1 could produce large fluorescence increase with PC6.

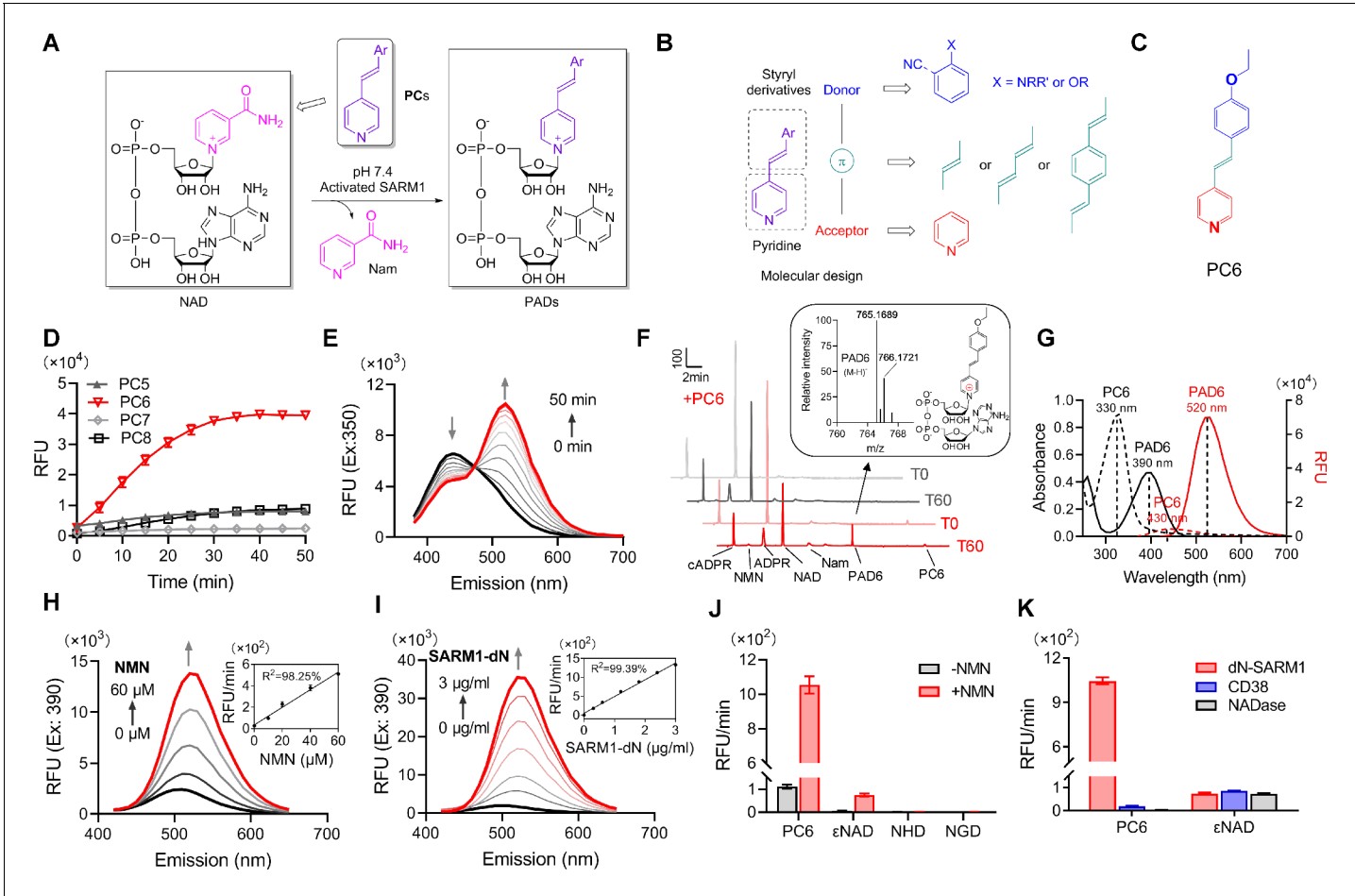

**Figure 1.** Design and characterization of PC probes. (**A**) Strategy of fluorescent imaging of the activated SARM1. (**B**) Designing based on pyridine and styryl derivatives with a donor-π-acceptor framework. (**C**) Structure of PC6. (**D**) The kinetics of the fluorescence increase at the maximal absorbance wavelengths catalyzed by SARM1-dN, in the presence of 100 μM NMN, 100 μM NAD, and 50 μM PCs. (**E**) Time-dependent changes of the emission spectra at the isosbestic point (350 nm). (**F**) HPLC analysis of PC6 reaction. Red line: in the presence of PC6, NMN, and NAD; Gray line: without PC6. Insert: MS analysis and structure of PAD6. (**G**) Absorbance and fluorescence spectra of 25 μM PC6/PAD6. (**H**) Emission spectra with dose of NMN (10, 20, 40, 60 μM) in the presence of NAD, PC6, and SARM1-dN. Inset: the initial rates plotted to NMN concentrations. (**I**) Emission spectra with doses of SARM1-dN in the presence of NMN, NAD, and PC6; Inset: the initial rate plotted to SARM1 concentrations. (**J**) The reaction rates of 10 μM PC6, in the presence of 100 μM NMN and 100 μM NAD, compared with NAD analogss (100 μM) catalyzed by SARM1. (**K**) The reaction rates of 10 μM PC6 catalyzed by SARM1, NADase, and CD38. PC = pyridine conjugate, NMN = nicotinamide mononucleotide.

The online version of this article includes the following source data and figure supplement(s) for figure 1:

**Source data 1.** Source data in excel for *Figure 1D-K*.

**Source data 2.** Source data in excel for *Figure 1—figure supplements 11C, 12* and *Figure 1—figure supplement 13*.

**Figure supplement 1.** Synthetic scheme of PC1–PC9. PC = pyridine conjugate.

**Figure supplement 2.** $^1$H NMR and $^{13}$C NMR spectra of **PC1** in DMSO-$d_6$.

**Figure supplement 3.** $^1$H NMR and $^{13}$C NMR spectra of **PC2** in CDCl$_3$.

**Figure supplement 4.** $^1$H NMR and $^{13}$C NMR spectra of **PC3** in CDCl$_3$.

**Figure supplement 5.** $^1$H NMR and $^{13}$C NMR spectra of **PC4** in CDCl$_3$.

**Figure supplement 6.** $^1$H NMR and $^{13}$C NMR spectra of **PC5** in DMSO-$d_6$.

**Figure supplement 7.** $^1$H NMR and $^{13}$C NMR spectra of **PC6** in CDCl$_3$.

**Figure supplement 8.** $^1$H NMR and $^{13}$C NMR spectra of **PC7** in CDCl$_3$.

**Figure supplement 9.** $^1$H NMR and $^{13}$C NMR spectra of **PC8** in CDCl$_3$.

**Figure supplement 10.** $^1$H NMR and $^{13}$C NMR spectra of **PC9** in CDCl$_3$.

**Figure supplement 11.** Structures of PC1–9 and activity screening.

**Figure supplement 12.** UV-vis absorption spectra scanning of the reactants.

**Figure supplement 13.** Fluorescence spectra of the reactants.

## Imaging SARM1 activation in live cells

PC6 was added to HEK293 cells overexpressing either wildtype SARM1 or the enzymatically inactive mutant, E642A (*Essuman et al., 2017*; *Zhao et al., 2019*; *Figure 2A*). Green fluorescence was clearly seen evenly distributed in the whole cells in the wildtype, but not in the mutant cells (*Figure 2B*), indicating active SARM1 was required. Intracellular production of PAD6 was confirmed in extracts of wildtype but not the E642A cells (*Figure 2C*). CZ-48, a cell-permeant mimetic of NMN and activator of SARM1 (*Zhao et al., 2019*), dramatically increased the PAD6 fluorescence (*Figure 2B*, right column) and none in E642A-cells. These results indicate that PC6 is cell-permeant and can be exchanged into the cytosolic NAD by the activated SARM1 to produce PAD6 having a large red shift in fluorescence. PAD6 was also cell-impermeant because of its charged ADP-ribose moiety and accumulated in the cytosol, greatly increased its detection sensitivity in live cells.

PC6 also could detect the activity of SARM1 endogenously expressed in HEK293T cells (*Zhao et al., 2019*). CZ-48 activated the endogenous SARM1 and produced increase of cytosolic PAD6 signal (*Figure 2D*, upper right), but none in the SARM1-knockout cells (*Figure 2D*, right lower), confirming the specificity of PC6 for SARM1.

An HEK293 cell line carrying doxycycline (Dox)-inducible SARM1 (*Zhao et al., 2019*) was used to further substantiate that the PAD6 fluorescence was derived from SARM1 activity. Without induction, only basal SARM1 (*Figure 2—figure supplement 1A*) with minimal activity was detected (*Figure 2E*, green dots), while activated by CZ-48, resulting in increase in PAD6-fluorescence (orange triangles). Induction of SARM1 (*Figure 2—figure supplement 1A*) produced minimal signal

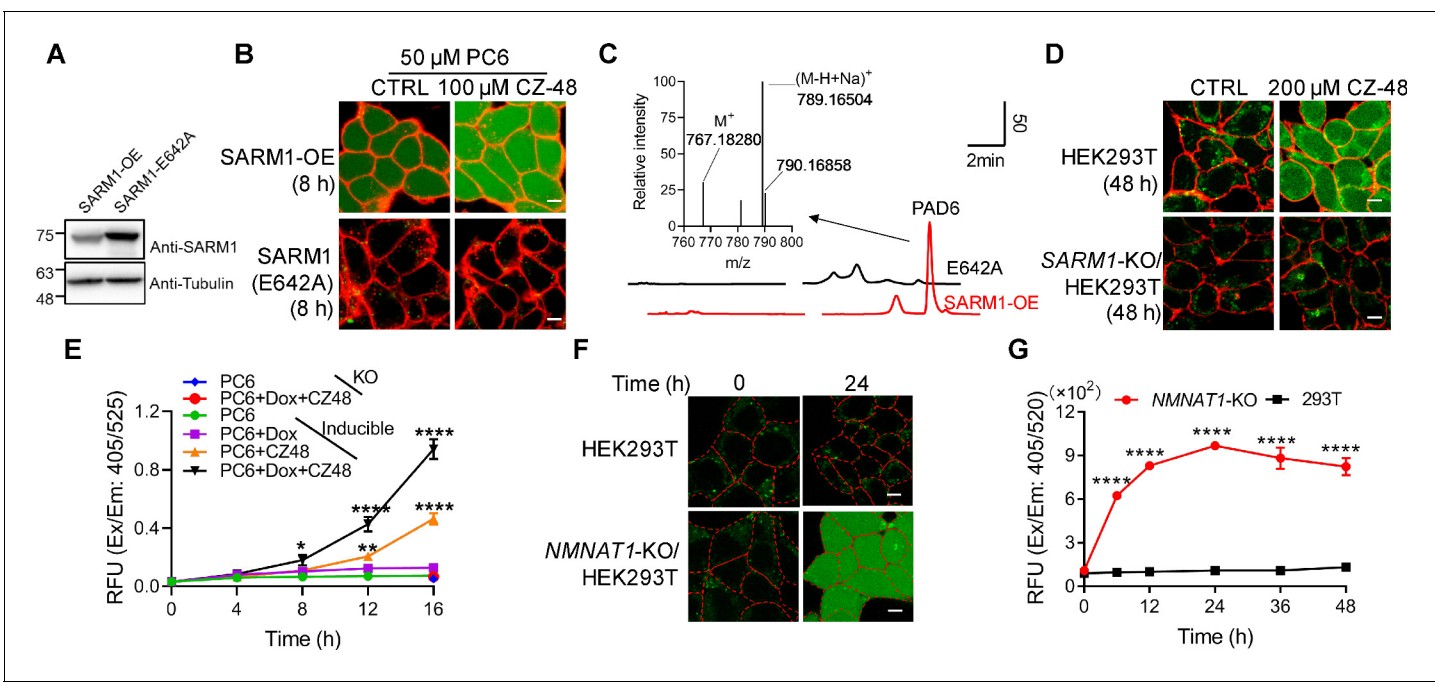

**Figure 2.** Live-cell imaging of SARM1 activation. (**A**) Western blot of the overexpression of SARM1 and inactive mutant, E642A in HEK293 cells. (**B**) Confocal fluorescence images of cells in (**A**) after incubation with PC6 in presence or absence of CZ-48. Green: PAD6; red: ConA-Alex-647; (**C**) HPLC and MS analysis of PAD6 from SARM1-OE cells. The metabolites were extracted by 0.6 M PCA from the cells in (**A**) after treating with 50 μM PC6 for 24 hr. Inset: MS analysis. (**D**) Confocal images of wildtype, or *SARM1*-KO HEK293T cells with PC6 in presence or absence of CZ-48. (**E**) The HEK293 cells carrying the inducible SARM1 were incubated with 50 μM PC6 in presence or absence of 0.5 mg/mL Dox and/or 100 μM CZ-48. The PAD6 fluorescence was analyzed by flow cytometry. (**F**) Confocal images of *NMNAT1*-KO/HEK293 T cells, incubated with PC6. Cell edges were marked according to the bright-field images. (**G**) Quantification of the cell fluorescence in (**F**). All the above experiments were repeated at least three times (means ± SDs; n = 3; Student's *t*-test, *p<0.05; **p<0.01, ****p<0.0001). Scale bar 10 μm.

The online version of this article includes the following source data and figure supplement(s) for figure 2:

**Source data 1.** Source data in excel for *Figure 2E and G*.

**Figure supplement 1.** Expression level of SARM1 for *Figure 2E* and the activities of CD38.

**Figure supplement 1—source data 1.** Source data in excel for *Figure 2—figure supplement 1B and C*.

also (*Figure 2E*, purple squares), confirming SARM1 is auto-inhibitory. With CZ-48, both the basal and the induced SARM1 were activated, resulting in the largest signal (*Figure 2E*, black triangles). In *SARM1*-knockout cells, no signal was detected (*Figure 2D*, *SARM1*-KO; *Figure 2E*, blue and red dots).

Endogenous NMN can be increased by ablating NMN-adenylyltransferase (NMNAT1) (*Zhao et al., 2019*) to activate SARM1. Correspondingly, *NMNAT1* knockout in HEK293T cells also resulted in increasing PAD6 fluorescence (*Figure 2F*) in a time-dependent manner (*Figure 2G*).

Consistent with the in vitro results showing that PC6 is highly selective for SARM1 over CD38 in live cells, cells expressing either wildtype or Type III mutant CD38 (*Liu et al., 2017*; *Zhao et al., 2012*) did not show PAD6-signal after 48 hr incubation with PC6 (*Figure 2—figure supplement 1B*),

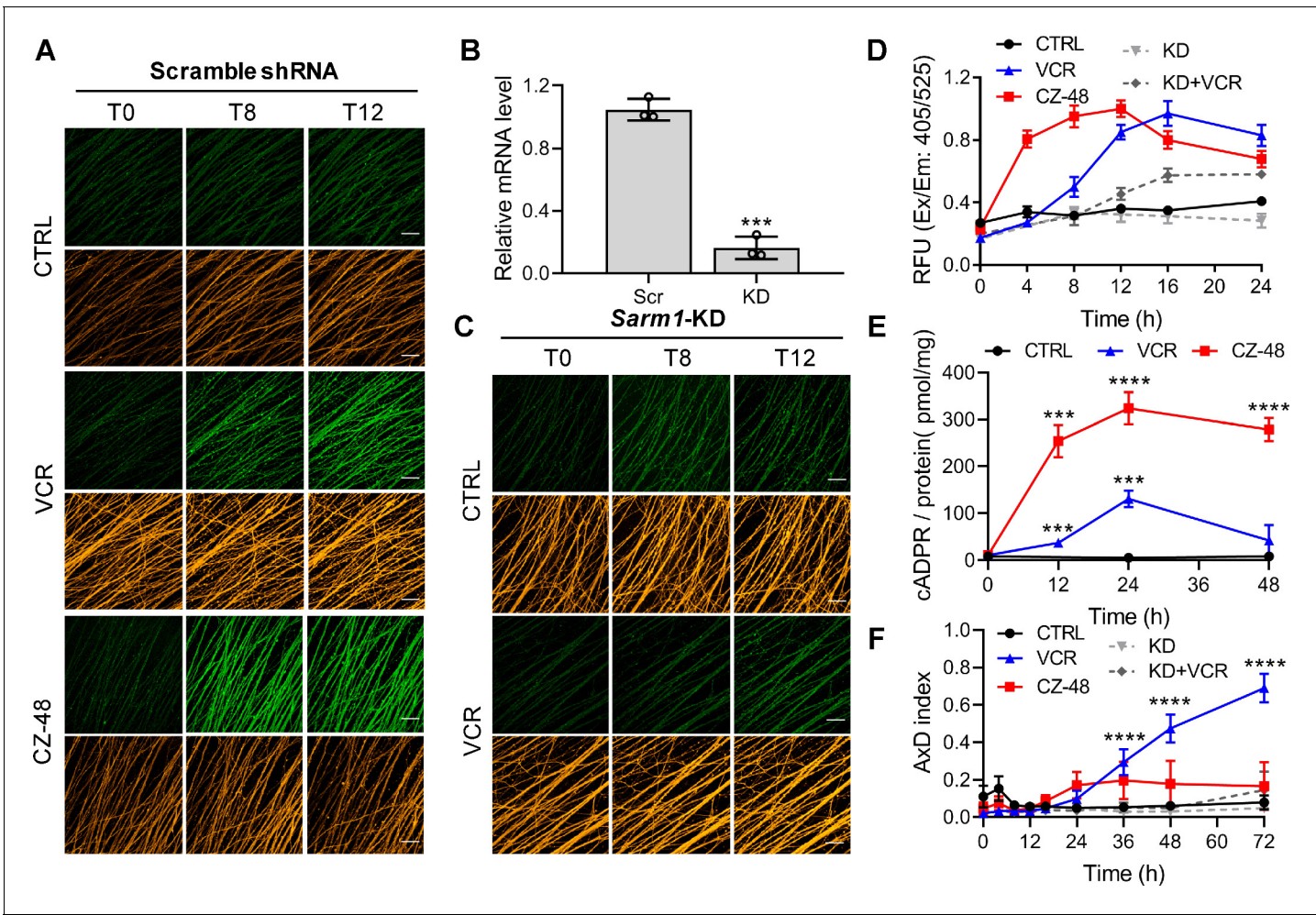

**Figure 3.** SARM1 activation in mouse DRG upon vincristine treatment. (**A, C**) Confocal imaging of SARM1 activation in DRG neuronal axons. The neurons were infected with virus expressing TdTomato to provide easy imaging of the axons. Cells were additionally transfected with either scramble (**A**) or *Sarm1*-specific (**C**) shRNAs and treated with 50 µM PC6, 200 µM CZ-48, or 50 nM Vincristine and imaged in the indicated time points. Green: PAD6; orange: TdTomato; scale bar 50 µm. (**B**) Knockdown efficiency of *Sarm1*. Scr, scramble shRNA; KD, *Sarm1*-specific shRNA. (**D**) Quantification of the fluorescence intensity of PAD6 in DRGs. (**E**) Intracellular cADPR contents. (**F**) Indices of AxD. All the above experiments were repeated at least three times (means ± SDs; n = 3; Student's *t*-test, ***p<0.001; ****p<0.0001). AxD = axon degeneration, cADPR = cyclic ADP-ribose, PC = pyridine conjugate, VCR = Vincristine.

The online version of this article includes the following source data and figure supplement(s) for figure 3:

**Source data 1.** Source data in excel for *Figure 3B, D-F*.

**Figure supplement 1.** Integrity of axons visualized by the TdTomato fluorescence.

even though the expressed enzymes readily increased cellular cADPR (*Figure 2—figure supplement 1C*).

## Imaging SARM1 activation during AxD

Vincristine (VCR)-induced AxD in peripheral neuropathy is a common side effect of chemotherapy (*Essuman et al., 2017*) and is thought to be due to SARM1-activation (*Gerdts et al., 2013*). Mouse dorsal root ganglion (DRG) neurons were infected with lentivirus carrying TdTomato for visualizing the axons (*Figure 3A and C*, orange), and with either a non-targeting (*Figure 3A*) or *Sarm1*-specific shRNA (*Figure 3C*). In the non-targeting group, VCR elevated PAD6-fluorescence (*Figure 3A*, green), indicating activation of SARM1, by as early as 4–8 hr and reaching a maximum by 16 hr (*Figure 3A and D*, blue). AxD started at about 20 hr (*Figure 3F*, blue; *Figure 3—figure supplement 1A*), temporally consistent with a causal role for SARM1. Another measure of SARM1 activation is the elevation of cellular cADPR (*Zhao et al., 2019*), which occurred (*Figure 3E*, blue) by 12 hr, peaking at 24 hr. Neurons not treated with VCR showed neither SARM1-activation nor AxD (*Figure 3A, D,E,F* and *Figure 3—figure supplement 1A*, CTRL).

Reducing endogenous SARM1 using shRNA (*Figure 3B,D and F*, KD) reduced the PAD6 fluorescence without altering its peaking at 16 hr (*Figure 3C*; *3D*, KD +VCR) and reduced AxD (*Figure 3F*, KD +VCR; *Figure 3—figure supplement 1B*), further substantiating a causal role for SARM1. CZ-48 induced SARM1-activation more rapidly (*Figure 3A and D*, red) and elevated cADPR higher (*Figure 3E*, red), confirming its direct action. Intriguingly, CZ-48 did not induce massive AxD as VCR (*Figure 3F*, CZ-48; *S5A*). These results indicate SARM1-activation is a necessary and causal factor, but not a sufficient one for AxD. Other critical factors and downstream events of microtubular dysfunction might contribute to the degeneration.

## Dehydronitrosonisodipine (dHNN) is an inhibitor of SARM1 activation

Another prompt application of PC6 is library screening for inhibitors of SARM1. The feasibility was verified by measuring the $IC_{50}$ of a reported inhibitor of SARM1, nicotinamide (Nam) (*Essuman et al., 2017*). The measured $IC_{50}$ value of Nam was around 140 µM, which is consistent with the reported value (*Figure 4—figure supplement 1A*). Next, we utilized this assay to screen for SARM1 inhibitors. NMN-activated SARM1 was incubated with drugs of the library (*Figure 4A*) and its activity measured with PC6 in the presence of NAD (*Figure 1*). Out of 2015 drugs, 34 had more than 80% inhibition (*Figure 4B*), which were further tested for inhibition of the SARM1-NADase activity using HPLC. *Figure 4C* shows the plot the $IC_{50}$-values of these drugs measured with both the PC6 and the NADase/HPLC assays. Twenty-four drugs are in the middle sector, indicating they inhibited both reactions similarly. Two inhibited the PC6 activity five fold less than the NADase (*Figure 4C*, left sector), and eight in the right sector (seven have $IC_{50}$s higher than 40 µM) inhibited NADase less than the base-exchange. These remarkable differences underscore the importance of using more than one assay for drug screening (see Discussion).

In the middle sector is nisoldipine (NSDP), a calcium channel blocker having beneficial effects on neurodegenerative diseases. Peculiarly, its inhibition of SARM1 varied widely among batches. Investigations indicated the active compound was not NSDP but its derivative. *Figure 4D* shows fresh NSDP had an $IC_{50}$-value of about 150 µM (squares), but its potency increased 75-fold after exposure to UV (*Figure 4D*, triangles, $IC_{50} = 2.36 \pm 0.3$ µM). Also, fresh NSDP had an HPLC-elution peak at 12.2 min (*Figure 4—figure supplement 1B*), but was completely converted by UV to a compound having a peak at 9.8 min that strongly inhibited SARM1 (*Figure 4E*, red). HRMS showed that the active compound had a mass of 370.15205 Da (*Figure 4E*, inset) identical to a known derivative of NSDP, dehydronitrosonisoldipine (dHNN) (*Marinkovic et al., 2003*). The HPLC-elution profile of the active compound was also the same as dHNN (*Figure 4—figure supplement 1B*, purple line and green dash). Indeed, authentic dHNN was active and could not be further activated by UV (*Figure 4F*, red line and dash), which also indicates that it is photostable. Another derivative of NSDP, dehydronisoldipine (dHN, elution peak at 8.7 min, *Figure 4—figure supplement 1B*), showed no inhibition before or after UV (*Figure 4F*, black line and dash), indicating that the nitroso group is essential for the effect.

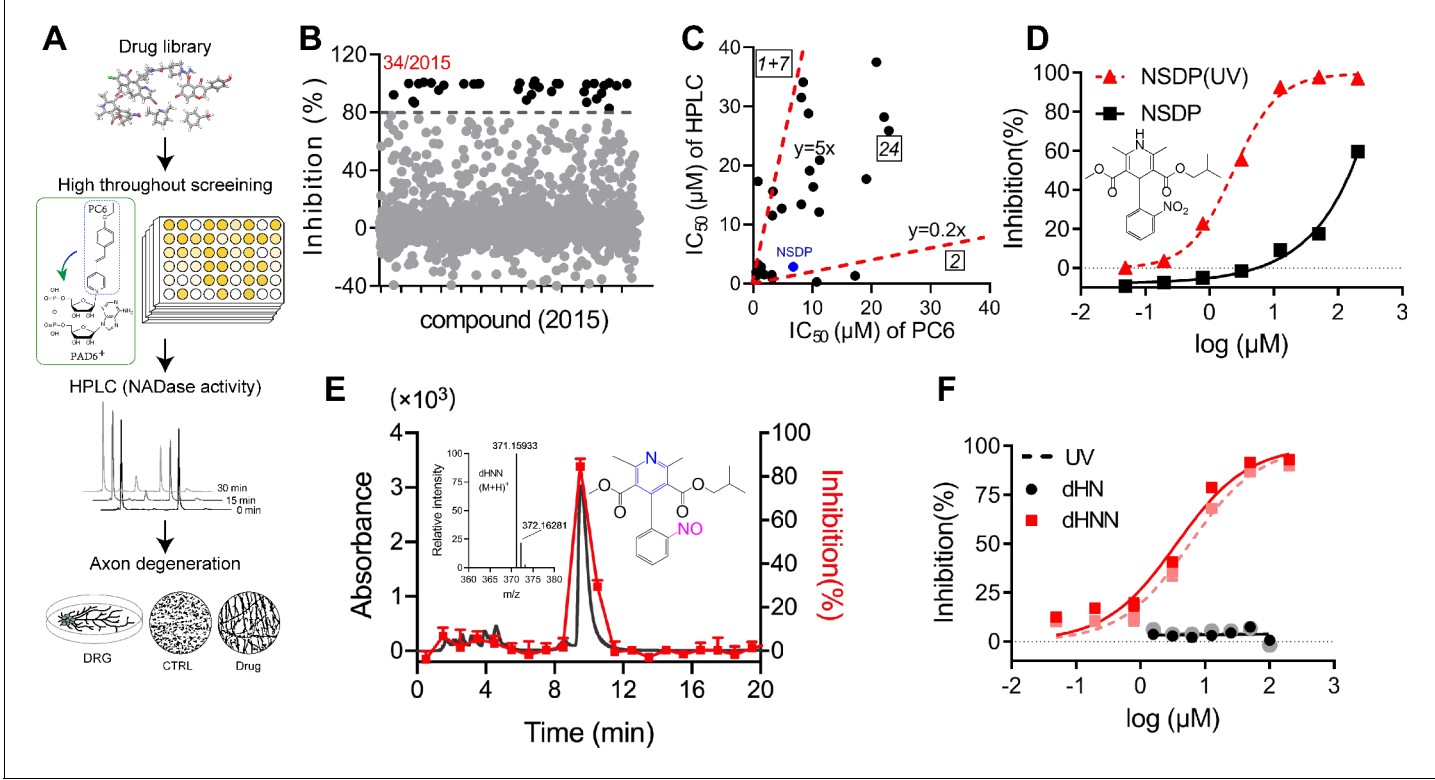

**Figure 4.** Identification of dHNN as an inhibitor of SARM1. (**A**) Flowchart of the PC6-based high-throughput screening. (**B**) Inhibitory effects of the 2015 compounds (50 µM) from an approval drug library. The activity of drug-treated SARM1-dN was determined with PC6 assay. The detail procedure and statistical analysis are referred to the 'Materials and methods' section. (**C**) Plot of IC$_{50}$s of the 27 most potent inhibitory compounds from high-throughput screening, determined by PC6 (x axis) versus by HPLC (y axis, NADase activity) assays. See 'Materials and methods' section. (**D**) Inhibition curves of NSDP before (black) and after (NSDP[UV], red) UV at 254 nm for 30 min. (**E**) HPLC elution profile of dHNN. NSDP after 30 min UV treatment was analyzed using a C-18 column with a gradient of 0.1% TFA and ACN in 0.1% TFA. Fractions were assayed for inhibition of SARM1-dN by PC6 assay. The derivative in the elution peak was identified by MS. Black line: absorbance at 275 nm; red dots: inhibition activity. Insets: MS of the peak fraction showing its mass was the same as dHNN and the chemical structure of dHNN. (**F**) Concentration-inhibition curves of dHN (black solid line), UV-treated dHN (black dotted line), dHNN (red solid line) and UV-treated dHNN (red dotted line), measured by PC6 assay. PC = pyridine conjugate, NSDP = nisoldipine .

The online version of this article includes the following source data and figure supplement(s) for figure 4:

**Source data 1.** Source data in excel for *Figure 4B-F*.
**Figure supplement 1.** Inhibitory mechanism of dHNN against SARM1.
**Figure supplement 1—source data 1.** Source data in excel for *Figure 4—figure supplement 1A-D*.

## dHNN inhibits SARM1 and AxD by covalently modifying cysteines

The dHNN-induced inhibition of SARM1 was irreversible by washing (*Figure 4—figure supplement 1C*, red bars), while that by Nam was reversible. Also, dHNN-inhibition was time-dependent, but not Nam (*Figure 4—figure supplement 1D*), strongly suggesting dHNN covalently reacted with SARM1.

To determine the mechanism of action of dHNN, we truncated the inhibitory ARM-domain, producing a constitutively active SAM-TIR, which showed a right-shifted inhibition curve compared to SARM1-dN (*Figure 5A*), with around 50-fold increase in the IC$_{50}$. The IC$_{50}$ of dHNN in the SARM1-dN-expression cells is around 4 µM, close to the IC$_{50}$ in vitro. dHNN decreased the cellular cADPR in cells expressing SARM1, but not in those expressing SAM-TIR (*Figure 5B*). These results suggest that dHNN is cell-permeant and acts mainly by blocking SARM1 activation but not its enzymatic activities.

The nitroso group of dHNN may covalently modify cysteine residues (*Callan et al., 2009*) in SARM1. Indeed, LC-MS/MS identified several dHNN-modified peptides, among which Cys311 in the ARM domain is the dominant one (*Figure 5C*, *Figure 5—figure supplement 1A–B*). Many peptides

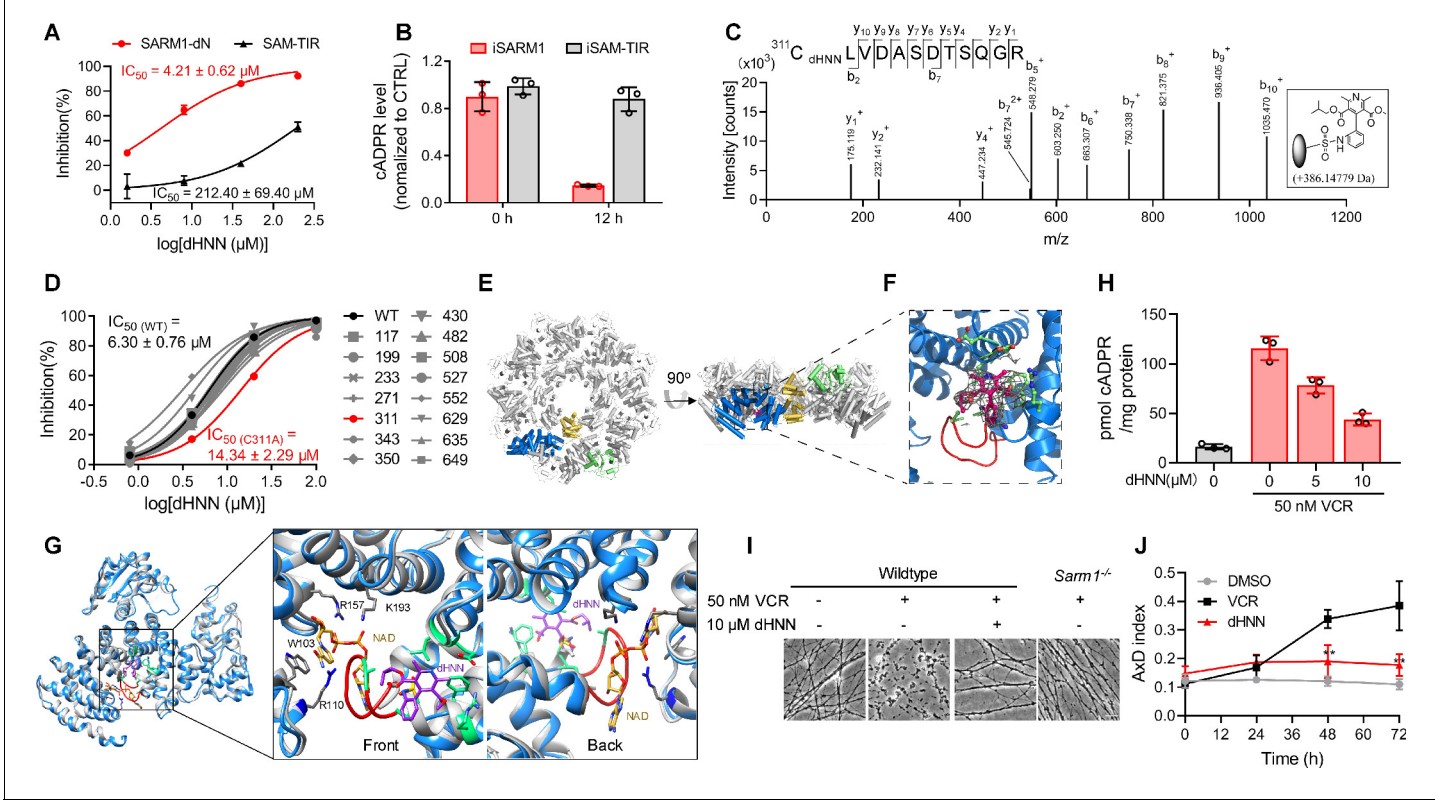

**Figure 5.** dHNN reduces AxD by inhibiting SARM1 through covalent modification of the cysteines. (**A**) Inhibition of SARM1-dN and SAM-TIR by dHNN in vitro. See 'Materials and methods'. (**B**) Inhibition of SARM1-dN and SAM-TIR by dHNN in cellulo. See 'Materials and methods'. (**C**) MS of SARM1-dN modification by dHNN. Peptide spectrum match shows that Cys311 was modified by dHNN, increasing its mass to 386.14779 Da. (**D**) Each cysteine in SARM1-dN was mutated to alanine. The dHNN-IC$_{50}$s were measured by PC6 assay. (**E**) Top (left) and side (right) view of the SARM1 octamer. α-helices are shown as cylinders. dHNN modifications are shown as sticks. One protomer is colored in blue for ARM, gold for SAM, and green for TIR. The other protomers are colored in grey. (**F**) Zoom-in view of the dHNN-modified pocket. dHNN: purple stick; the electron density corresponding to dHNN: grey mesh; interacting residues: green; loop: red. (**G**) Superposition of SARM1-dHNN onto SARM1-NAD (PDB: 7mc6). SARM1-dHNN and SARM1-NAD were shown as blue and grey cartoon, respectively. The insertion loop in SARM1-dHNN was shown in red. NAD was shown as stick models and colored with gold carbons. Residues interacting with NAD, W103, R110 and R157, were shown as stick models with grey carbons. dHNN and residues interacting with dHNN were shown as in (**F**). (**H**) DRG neurons were treated with dHNN for 16 hr in the presence of VCR. The cellular cADPR contents were measured. (**I**) Micrographs of AxD of DRG neurons after treatment of VCR in presence of dHNN for 72 hr. (**J**) Quantification of AxD indices after 0, 24, 48, 72 hr treatment with VCR as in (**I**). AxD = axon degeneration. VCR = Vincristine.

The online version of this article includes the following source data and figure supplement(s) for figure 5:

**Source data 1.** Source data in excel for *Figure 5A, B, D, H and J*.

**Figure supplement 1.** dHNN modifications on the peptides of SARM1 or nonspecific proteins analyzed by LC-MS/MS.

**Figure supplement 1—source data 1.** Source data in excel of *Figure 5—figure supplement 1A–C*.

**Figure supplement 2.** Data processing procedure for the SARM1-dHNN structure.

**Figure supplement 3.** Structure of SARM1 was stabled in inactive form after dHNN treatment.

**Figure supplement 4.** dHNN attenuated the axotomy-induced AxD.

of other proteins with cysteines were also identified but none showed modification by dHNN (*Figure 5—figure supplement 1C*), indicating specificity of dHNN. Single mutation of all the cysteines to alanine showed that C311A significantly decreased the response to dHNN (*Figure 5D*). However, the IC$_{50}$ of C311A was only two fold higher than that of the wildtype, which indicates dHNN might modify other cysteines when Cys311 is mutated and inactivate SARM1.

With Cryo-EM, we found that dHNN stabilized a similar inhibitory conformation of SARM1 as that induced by NAD (PDB: 7cm6) (*Jiang et al., 2020*). In 2D-classification of the untreated SARM1, most particles presented only the SAM octamer ring (*Figure 5—figure supplement 3A*). For the dHNN-treated SARM1, larger octamer ring corresponding to both the SAM and ARM/TIR domains could

be clearly observed (*Figure 5—figure supplement 2*, *Supplementary file 1*, *Figure 5—figure supplement 3B*). Its 3D-structure was constructed at 2.4 Å resolution (*Figure 5—figure supplement 3C*) with residues from 56 to 549 (ARM and SAM domains) and 561 to 702 (TIR domain) all fitted into the cryo-EM map (*Figure 5E* and *Figure 5—figure supplement 3D*). Structural superimposition with the NAD-bound SARM1 (PDB: 7cm6) (*Jiang et al., 2020*) showed RMSD values of 0.91 (*Figure 5—figure supplement 3E*), suggesting that dHNN constrains SARM1 in an inactive conformation similar to that induced by NAD. Extra electron density was only observed near residue Cys311, the dHNN-target (*Figure 5C and D*), but not other cysteine residues, consistent with it being derived from dHNN (5F, purple). dHNN interacts with Glu264, Leu268, Arg307, Phe308, and Ala315 (*Figure 5F* and *Figure 5G*, green) in the ARM domain, pushing the insertion loop (*Figure 5F*, red) toward ARM1 and stabilizes the domain. This is similar to that observed with NAD, which binds at the other side of the insertion loop and stabilizes the ARM domain possibly via ligating ARM1 and the insertion loop (*Figure 5G*).

By preventing SARM1 from activation, dHNN also inhibited the VCR-activated cADPR production (*Figure 5H*) in neurons and blocked not only the VCR-induced AxD (*Figure 5I*, third picture; *Figure 5J*, red line), as effective as knocking out SARM1 (*Figure 5I*, fourth picture), but also AxD after axotomy (*Figure 5—figure supplement 4A*, third column; *B*, red line).

## Discussion

Visualizing the activity of a signaling enzyme in live cells provides clearer understanding of the spatial and temporal aspects of its mechanism and function, a goal sought by many. The PC probes developed here are particularly advantageous. They are cell-permeant, but the SARM1-catalyzed exchange products are not and accumulate in the cytosol, enhancing their detection. The remarkably large red shift of the product fluorescence provides easy visualization away from the interference of autofluorescence.

Currently, there are several fluorescent probes for SARM1 activity in use. They are all analogs of NAD and are cell-impermeant, such as εNAD. The fact that NAD is now shown to be an inhibitor of SARM1 (*Jiang et al., 2020*; *Sporny et al., 2020*) makes the use of these analog probes problematic, as they may affect SARM1 activity as well. PC6 has no such drawback as it is a pyridine, not an NAD analog. Furthermore, the use of PC6 requires neither expression of construct nor cell manipulation, making it suitable for detecting SARM1 activity in any cells. This is documented in this study. Using CZ-48, a cell-permeant activator, to activate endogenous SARM1 produces large increase in PAD6 fluorescence in a variety cell lines as well as primary neurons. With the probes, we provided the first direct evidence in live DRG neurons that SARM1 activation precedes AxD by several hours and that it is a necessary but insufficient factor for AxD.

Screening library to identify compounds of interest is a straightforward strategy widely used. The case for SARM1 is more complicated, as it is not only a multi-domain protein but also an auto-regulated enzyme catalyzing multiple reactions. Compounds may target the regulatory ARM domain as shown here for dHNN, or the catalytic TIR domain as Nam and the inhibitors reported during the preparation of this manuscript (*Hughes et al., 2021*; *Loring et al., 2020*). For SARM1, the substrates are different for the base-exchange and the NADase reactions and may thus be differentially affected by the inhibitor-induced conformational changes of the catalytic site. Although the exact reason remains to be determined, the compounds shown here that can selectively block one reaction much more than the other are of interest. Many believe that the NADase activity of SARM1, leading to cellular NAD depletion, is its dominant property for regulating AxD. But the two calcium messengers, cADPR and NAADP produced by its cyclizing and base-exchange reactions may well have functional roles as well. Compounds with differential inhibition can thus be an important tool to resolve the issue.

Much effort is being invested in targeting SARM1-mediated NAD depletion for therapeutic protection from AxD. Chemical blockers may well be an ideal tool for turning off the NAD depletion. dHNN uncovered in this study is the first compound ever described that can block the activation of SARM1, revealing a druggable allosteric site and can thus usher in a new approach for therapeutic drug development. Another point of interest is that dHNN is a derivative of NSDP. Metabolic conversion of NSDP to dHNN, leading to inhibition of SARM1, may well account for the neural protective effects of NSDP (*Siddiqi et al., 2019*).

## Materials and methods

### Animals

This study was carried out in strict accordance with animal use protocol approved by Peking University Shenzhen Graduate School Animal Care and Use Committee (#AP0015001). All animals (C57BL6/J), purchased from Guangdong Medical Laboratory Animal Center (China), were handled in accordance with the guidelines of the Committee on the Ethic of Animal Experiments. All surgery was performed after euthanasia and efforts were made to minimize suffering.

### Cell lines

The HEK293 and HEK293T cells were purchased from the American Type Culture Collection and the identity has been authenticated by STR profiling. They have not been contaminated by mycoplasma. The cells were cultured in DMEM supplemented with 10% fetal calf serum and 1% penicillin-streptomycin solution and maintained in a standard humidified tissue culture incubator with 5% $CO_2$.

### Reagents

NAD, NMN, Digitonin, Poly-L-lysine, 5-fluoro-2'-deoxyuridine and uridine, $KH_2PO_4$, $NH_4HCO_3$, and iodoacetic acid were purchased from Sigma-Aldrich. DMEM, Neurobasal Plus Medium, Trypsin-EDTA, penicillin/streptomycin solution, B27 plus, GlutaMax, Laminin, Lipofectamine 2000, ConA-Alex-647, formic acid, acetonitrile were purchased from ThermoFisher. NGF was from Sino Biological. FBS was obtained from PAN Bitotech. Approval drug library (L1000) and Nisoldipine power (Cas # 63675-72-9) were purchased from TargetMol. Dehydro Nisoldipine (Cas #103026-83-1) was obtained from TRC, while dehydronitrosonisoldipine (Cas #87375-91-5) was purchased from Glpbio and TRC. Vincristine was purchased from Selleck. General chemicals for probe synthesis were purchased from Dieckmann, Alfa, Energy, or Sangon Biotech (Shanghai).

### Synthesis and characterization of pyridine conjugates (PCs)

All air and water-sensitive reactions were carried out with anhydrous solvents in flame-dried flasks under argon atmosphere, unless otherwise specified. All the reagents were obtained commercially and used without further purification, unless otherwise specified. Anhydrous DMF was vacuum distilled from barium oxide, acetonitrile, and dichloromethane was distilled from calcium hydride. Yields refer to isolated yields, unless otherwise specified. Reactions were monitored by thin-layer chromatography (TLC) carried out on 0.25 mm silica gel plates (60 F-254) that were analyzed by UV light as visualizing method and by staining with anisaldehyde (450 mL of 95% EtOH, 25 mL of conc. $H_2SO_4$, 15 mL of acetic acid, and 25 mL of anisaldehyde) or $KMnO_4$ (200 mL $H_2O$ of 1.5 g $KMnO_4$, 10 g $K_2CO_3$ and 1.25 mL of 10% aq. NaOH). Silica gel (200–300 mesh) was used for flash column chromatography. Nuclear magnetic resonance (NMR) spectra were recorded on either a 300 ($^1$H: 300 MHz, $^{13}$C: 75 MHz), 400 ($^1$H: 400 MHz, $^{13}$C: 100 MHz), or 500 ($^1$H: 500 MHz, $^{13}$C: 125 MHz) NMR spectrometer. The following abbreviations were used to explain the multiplicities: s = singlet, d = doublet, t = triplet, q = quartet, dd = doublet of doublets, m = multiplet, br = broad. High resolution mass spectra (HRMS) were obtained from a MALDI-TOF mass spectrometer.

To synthesize PC1, to a stirred solution of 4-vinylpyridine (210 mg, 2.0 mmol), 4-iodoaniline (220 mg, 1.0 mmol), P(o-tol)$_3$ (61 mg, 20 mol%), and triethylamine (0.40 mL, 2.9 mmol) in degassed $CH_3CN$ (15 mL) under argon was added Pd(OAc)$_2$ (23 mg, 10 mol%) quickly. The resulting mixture was stirred at 100°C for 5 hr. The mixture was then diluted with water (20 mL) and aqueous phase was extracted with ethyl acetate (15 mL ×3). The combined organic extracts were dried over anhydrous $Na_2SO_4$, filtered and evaporated under reduced pressure. Silica gel flash column chromatography (ethyl acetate/hexanes = 3:1) of the residue gave a pale-yellow solid (66 mg, 34%) as the product. PC1: mp = 272–273°C. $^1$H NMR (400 MHz, DMSO-$d_6$) δ 8.47 (d, $J$ = 5.3 Hz, 2H), 7.46 (d, $J$ = 6.0 Hz, 2H), 7.42–7.31 (m, 3H), 6.88 (d, $J$ = 16.4 Hz, 1H), 6.59 (d, $J$ = 8.5 Hz, 2H), 5.51 (s, 2H). $^{13}$C NMR (100 MHz, DMSO-$d_6$) δ 149.8, 145.4, 133.7, 128.5, 123.6, 120.0, 120.0, 113.8. HRMS (+ESI) m/z calcd. for $C_{13}H_{12}N_2$ (M + H)$^+$ 197.1073, found 197.1072.

To synthesize PC2, to a stirred solution of 4-iodoaniline (0.60 g, 2.7 mmol) in DMF (7.5 mL) was added ethyl bromide (0.25 mL, 3.35 mmol) and $Na_2CO_3$ (0.50 g, 4.72 mmol) at rt. The resulting mixture was stirred at 70°C for 6 hr. The mixture was then diluted with water (20 mL) and the aqueous

phase was extracted with ethyl acetate (15 mL ×3). The combined organic extracts were washed with water (15 mL ×3), dried over anhydrous $Na_2SO_4$, filtered and evaporated under reduced pressure. Silica gel flash column chromatography (ethyl acetate/hexanes = 1:20) gave a brown solid (79 mg, 12%) as the product (**1**). Spectral data of **1** are consistent with those reported in the literature. (*Ni et al., 2017*) To a stirred solution of 4-vinlypridine (53 mg, 0.5 mmol), **1** (74 mg, 0.3 mmol), P(*o*-tol)$_3$ (30 mg, 20 mol%), and triethylamine (0.40 mL, 2.9 mmol) in degassed $CH_3CN$ (5 mL) under argon was added Pd(OAc)$_2$ (11 mg, 10 mol%) quickly. The resulting mixture was stirred at 100°C for 12 hr. The mixture was then diluted with water (20 mL) and aqueous phase was extracted with ethyl acetate (15 mL ×3). The combined organic extracts were dried over anhydrous $Na_2SO_4$, filtered and evaporated under reduced pressure. Silica gel flash column chromatography (ethyl acetate/hexanes = 2:1) of the residue gave a pale-orange solid (51 mg, 76%) as the product. PC2: mp = 199–200°C. $^1H$ NMR (400 MHz, $CDCl_3$) δ 8.52 (d, *J* = 5.8 Hz, 2H), 7.39 (d, *J* = 8.5 Hz, 2H), 7.31 (dd, *J* = 4.7, 1.4 Hz, 2H), 7.23 (d, *J* = 16.3 Hz, 1H), 6.79 (d, *J* = 16.2 Hz, 1H), 6.60 (d, *J* = 8.6 Hz, 2H), 3.21 (q, *J* = 7.1 Hz, 2H), 1.29 (t, *J* = 7.2 Hz, 3H). $^{13}C$ NMR (125 MHz, $CDCl_3$) δ 149.9, 149.0, 145.5, 133.4, 128.5, 125.2, 121.2, 120.4, 112.6, 38.2, 14.8. HRMS (+ESI) m/z calcd. for $C_{15}H_{16}N_2$ (M + H)$^+$ 224.1386, found 224.1382.

To synthesize PC3, to a stirred solution of 4-iodoaniline (1.2 g, 5.5 mmol) in DMF (15 mL) was added ethyl bromide (2.0 mL, 27 mmol) and $Na_2CO_3$ (1.0 g, 9.4 mmol) at rt. The resulting mixture was stirred at 70°C for 6 hr. The mixture was then diluted with water (20 mL) and the aqueous phase was extracted with ethyl acetate (15 mL ×3). The combined organic extracts were washed with water (15 mL ×3), dried over anhydrous $Na_2SO_4$, filtered, and evaporated under reduced pressure. Silica gel flash column chromatography of the residue (ethyl acetate/hexanes = 1: 30) gave a brown oil (814 mg, 54%) as the product (**2**). Spectral data of **2** is consistent with those reported.(*Kolvari et al., 2014*) To a stirred solution of **2** (273 mg, 1.0 mmol), 4-vinylpyridine (210 mg, 2.0 mmol), P(*o*-tol)$_3$ (61 mg, 20 mol%), and triethylamine (0.40 mL, 2.9 mmol) in degassed $CH_3CN$ (15 mL) under argon was added Pd(OAc)$_2$ (23 mg, 10 mol%) quickly. The resulting mixture was stirred at 100°C for 12 hr. The mixture then was diluted with water (20 mL) and the aqueous phase extracted with ethyl acetate (15 mL ×3). The combined organic extracts were dried over anhydrous $Na_2SO_4$, filtered, and evaporated under reduced pressure. Silica gel flash column chromatography (ethyl acetate/hexanes = 1:1) of the residue gave a pale-yellow solid (138 mg, 55%) as the product. **PC3**: mp = 184–185°C. $^1H$ NMR (400 MHz, $CDCl_3$) δ 8.50 (d, *J* = 5.6 Hz, 2H), 7.47–7.38 (m, 2H), 7.30 (d, *J* = 6.1 Hz, 2H), 7.25–7.19 (m, 1H), 6.74 (t, *J* = 16.9 Hz, 1H), 6.66 (t, *J* = 5.9 Hz, 2H), 3.47–3.18 (m, 4H), 1.31–1.02 (m, 6H). $^{13}C$ NMR (100 MHz, $CDCl_3$) δ 149.9, 148.2, 145.7, 133.4, 128.6, 123.2, 120.4, 120.4, 111.5, 44.5, 12.7. HRMS (+ESI) m/z calcd. for $C_{17}H_{20}N_2$ (M + H)$^+$ 253.1699, found 253.1699.

To synthesize PC4, to a stirred solution of **2** (273 mg, 1.0 mmol), 1-bromo-4-vinylbenzene (183 mg, 1.0 mmol), P(*o*-tol)$_3$ (61 mg, 20 mol%), triethylamine (0.40 mL, 2.9 mmol) in degassed $CH_3CN$ (15 mL) under argon was added Pd(OAc)$_2$ (23 mg, 10 mol%) quickly. The resulting mixture was stirred at 100°C for 12 hr. The mixture was then diluted with water (20 mL) and the aqueous phase was extracted with ethyl acetate (15 mL ×3). The combined organic extracts were dried over anhydrous $Na_2SO_4$, filtered, and evaporated under reduced pressure. Silica flash column chromatography (ethyl acetate/hexanes = 1:30) gave a pale green solid (234 mg, 71%) as the product (**3**). Spectral data of **3** are consistent with those reported in the literature.(*Lemercier et al., 2006*) To a stirred solution of **3** (165 mg, 0.50 mmol), 4-vinlypyridine (105 mg, 1.0 mmol), P(*o*-tol)$_3$ (30 mg, 20 mol%), and triethylamine (0.20 mL, 1.5 mmol) in degassed $CH_3CN$ (10 mL) under argon was added Pd(OAc)$_2$ (11 mg, 10%) quickly. The resulting mixture was stirred at 100°C for 12 hr. The mixture was then diluted with water (20 mL) and the aqueous phase was extracted with ethyl acetate (15 mL ×3). The combined organic extracts were dried over anhydrous $Na_2SO_4$, filtered and evaporated under reduced pressure. Silica gel flash column chromatography (ethyl acetate/hexanes = 3:1) of the residue gave a pale-yellow solid (128 mg, 72%) as the product. PC4: mp = 225-226 °C. $^1H$ NMR (400 MHz, $CDCl_3$) δ 8.54 (d, *J* = 5.7 Hz, 2H), 7.54–7.44 (m, 4H), 7.42–7.38 (m, 2H), 7.36 (d, *J* = 6.0 Hz, 2H), 7.29 (d, *J* = 16.3 Hz, 1H), 7.09 (d, *J* = 16.2 Hz, 1H), 6.99 (d, *J* = 16.3 Hz, 1H), 6.88 (d, *J* = 16.2 Hz, 1H), 6.67 (d, *J* = 8.9 Hz, 2H), 3.39 (q, *J* = 7.0 Hz, 4H), 1.19 (t, *J* = 7.0 Hz, 6H). $^{13}C$ NMR (100 MHz, $CDCl_3$) δ 150.1, 147.6, 144.9, 139.1, 134.2, 133.1, 129.7, 129.0, 128.6, 128.0, 127.4, 126.3, 124.9, 124.4, 122.9, 120.8, 111.7, 44.4, 12.7. HRMS (+ESI) m/z calcd. for $C_{25}H_{26}N_2$ (M + H)$^+$ 355.2169, found 355.2167.

To synthesize PC5, to a stirred solution of 4-vinylpyridine (210 mg, 2.0 mmol), 4-iodophenol (220 mg, 1.0 mmol), P(o-tol)$_3$ (61 mg, 20 mol%), and triethylamine (0.40 mL, 2.9 mmol) in degassed CH$_3$CN (15 mL) under argon was added Pd(OAc)$_2$ (23 mg, 10 mol%) quickly. The resulting mixture was stirred at 100°C for 12 hr. The mixture was then diluted with water (20 mL). Upon addition of 5% HCl leads to partial precipitation of the product. The aqueous phase was extracted with ethyl acetate (15 mL ×3). The combined organic extracts were dried over anhydrous Na$_2$SO$_4$, filtered and evaporated under reduced pressure. Silica gel flash column chromatography (ethyl acetate/hexanes = 3:1) of the residue gave an off-white solid (130 mg, 66%) as the product. **PC5**: mp = 281–282. $^1$H NMR (400 MHz, DMSO-$d_6$) δ 9.83 (s, 1H), 8.55 (d, $J$ = 4.5 Hz, 2H), 7.67–7.42 (m, 5H), 7.07 (d, $J$ = 16.4 Hz, 1H), 6.85 (d, $J$ = 9.3 Hz, 2H). $^{13}$C NMR (100 MHz, DMSO-$d_6$) δ 158.2, 149.8, 144.8, 133.1, 128.7, 127.2, 122.4, 120.5, 115.7. HRMS (+ESI) m/z calcd. for C$_{13}$H$_{11}$NO (M + H)$^+$ 198.0913, found 198.0913.

To synthesize PC6, to a stirred solution of 4-vinylpyridine (631 mg, 6.0 mmol), 1-ethoxy-4-iodobezene (1.24 g, 5.0 mmol), P(o-tol)$_3$ (305 mg, 20 mol%), and triethylamine (2.0 mL, 15 mmol) in degassed CH$_3$CN (15 mL) under argon was added Pd(OAc)$_2$ (112 mg, 10 mol%) quickly. The resulting mixture was stirred at 100°C for 12 hr. The mixture was then diluted with water (20 mL) and the aqueous phase was extracted with ethyl acetate (15 mL ×3). The combined organic extracts were dried over anhydrous Na$_2$SO$_4$, filtered, and evaporated under reduced pressure. Silica gel flash column chromatography (ethyl acetate/hexanes = 3:1) of the residue gave an off-white solid (958 mg, 85%) as the product. PC6: mp = 146–147°C. $^1$H NMR (400 MHz, CDCl$_3$) δ 8.55 (dd, $J$ = 4.6, 1.5 Hz, 2H), 7.51–7.42 (m, 2H), 7.33 (dd, $J$ = 4.6, 1.5 Hz, 2H), 7.27 (t, $J$ = 8.1 Hz, 1H), 6.97–6.81 (m, 3H), 4.07 (q, $J$ = 7.0 Hz, 2H), 1.44 (t, $J$ = 7.0 Hz, 3H). $^{13}$C NMR (100 MHz, CDCl$_3$) δ 159.6, 150.1, 145.0, 132.8, 128.7, 128.4, 123.6, 120.6, 114.8, 63.6, 14.8. HRMS (+ESI) m/z calcd. for C$_{15}$H$_{15}$NO (M + H)$^+$ 226.1226, found 226.1226.

To synthesize PC7, to a stirred solution of 4-iodophenol (1.09 g, 4.93 mmol) and triethylamine (749 mg, 7.40 mmol) in CH$_2$Cl$_2$ (25 mL) was added acetyl chloride (465 mg, 5.92 mmol) at rt. The resulting mixture was stirred at 0°C for 20 min and then rt for 2 hr. The solution was then diluted with water (20 mL) and the aqueous phase was extracted with ethyl acetate (15 mL ×3). The combined extracts were dried over anhydrous Na$_2$SO$_4$, filtered, and evaporated under reduced pressure. The resulting pale brown oil (1.15 g, 89%) was obtained as the product (**4**) and was used for the next step without any further manipulation. Spectral data of **4** is consistent with those reported in the literature (*Flaherty et al., 2010*). To a stirred solution of 4-vinylpyridine (210 mg, 2.0 mmol), 4 (240 mg, 1.0 mmol), P(o-tol)$_3$ (61 mg, 20 mol%), and triethylamine (0.40 mL, 2.9 mmol) in degassed CH$_3$CN (15 mL) under argon was added Pd(OAc)$_2$ (23 mg, 10 mol%) quickly. The resulting mixture was stirred at 100°C for 6 hr. The mixture was then diluted with water (30 mL) and the aqueous phase was extracted with ethyl acetate (15 mL ×3). The combined organic extracts were dried over anhydrous Na$_2$SO$_4$, filtered and evaporated under reduced pressure. Silica gel flash column chromatography (ethyl acetate/hexanes = 3:1) of the residue gave a white solid (103 mg, 43%) as the product. PC7: mp = 152–153°C. $^1$H NMR (400 MHz, CDCl$_3$) δ 8.58 (d, $J$ = 5.9 Hz, 2H), 7.63–7.48 (m, 2H), 7.36 (dd, $J$ = 4.7, 1.4 Hz, 2H), 7.28 (d, $J$ = 16.3 Hz, 1H), 7.15–7.09 (m, 2H), 6.97 (d, $J$ = 16.3 Hz, 1H), 2.32 (s, 3H). $^{13}$C NMR (100 MHz, CDCl$_3$) δ 169.4, 150.9, 150.2, 144.5, 133.9, 132.2, 128.1, 126.2, 122.1, 120.9, 21.2. HRMS (+ESI) m/z calcd. for C$_{15}$H$_{13}$NO$_2$ (M + H)$^+$ 240.1019, found 240.1018.

To synthesize PC8, to a stirred solution of 5-bromo-2-hydroxy-benzonitrile (0.60 g, 30 mmol) in DMF (7 mL) was added ethyl bromide (0.37 mL, 5.0 mmol), and K$_2$CO$_3$ (1.0 g, 9.4 mmol) at rt. The resulting mixture was stirred at 70°C for 6 hr. The mixture was then diluted with water (20 mL), and the aqueous phase was extracted with ethyl acetate (15 mL ×3). The combined organic extracts were washed with water (15 mL ×3), dried over anhydrous Na$_2$SO$_4$, and evaporated under reduced pressure. A white solid was obtained as the product (**12**). To a stirred solution of the crude product (**12**), 4-vinylpyridine (315 mg, 3.0 mmol), P(o-tol)$_3$ (183 mg, 20 mol%), and triethylamine (1.2 mL, 8.7 mmol) in degassed CH$_3$CN (30 mL) under argon was added Pd(OAc)$_2$ (67mg, 10 mol%) quickly. The resulting mixture was stirred at 100°C for 5 hr. The mixture was then diluted with water (30 mL) and the aqueous phase was extracted with ethyl acetate (15 mL ×3). The combined organic extracts were dried over anhydrous Na$_2$SO$_4$, filtered and evaporated under reduced pressure. Silica gel flash column chromatography (ethyl acetate/hexanes = 3:1) of the residue gave a pale-yellow solid (433 mg, 58%) as the product. **PC8**: mp = 114–115°C. $^1$H NMR (400 MHz, CDCl$_3$) δ 8.59 (d, $J$ = 5.9 Hz, 2H), 7.80–7.60 (m, 2H), 7.35 (d, $J$ = 5.9 Hz, 2H), 7.19 (d, $J$ = 16.3 Hz, 1H), 6.99 (d, $J$ = 8.8 Hz, 1H),

6.91 (d, $J$ = 16.3 Hz, 1H), 4.19 (q, $J$ = 7.0 Hz, 2H), 1.50 (t, $J$ = 7.0 Hz, 3H). $^{13}$C NMR (101 MHz, CDCl$_3$) δ 160.4, 150.0, 143.9, 132.5, 131.8, 130.2, 128.9, 125.7, 120.6, 115.9, 112.3, 102.4, 64.8, 14.3. HRMS (+ESI) m/z calcd. for C$_{16}$H$_{14}$N$_2$O (M + H)$^+$ 251.1179, found 251.1178.

To synthesize PC9, to a stirred solution of 1-ethoxy-4-iodobezene (1.24 g, 5.0 mmol) and 3,3-diethoxyprop-1-ene (1.03 g, 7.9 mmol), P(o-tol) (305 mg, 20 mol%), Cs$_2$CO$_3$ (2.28 g, 7.0 mmol) and KCl (370 mg, 5 mmol) in DMF (30 mL) under argon was added Pd(OAc)$_2$ (115 mg, 10 mol%) quickly. The resulting mixture was stirred at 90℃ for 5 hr and then treated with 5% HCl (10 mL) and stirred at rt for 10 min. The mixture was then diluted with water (20 mL) and the aqueous phase was extracted with ethyl acetate (15 mL ×3). The combined organic extracts were dried over anhydrous Na$_2$SO$_4$, filtered and evaporated under reduced pressure. Silica gel flash column chromatography (ethyl acetate/hexanes = 1:5) gave a pale-yellow solid (443 mg, 50%) as the product (**6**). Spectral data of **6** is consistent with those reported in the literature (*Lator et al., 2018*). To a stirred solution of **6** (88 mg, 0.50 mmol), 4-methylpyridine (93 mg, 1.0 mmol) in Ac$_2$O (2 cmL) was added NaOAc (272 mg, 2.0 mmol) at rt. The resulting mixture was heated under reflux for 21 hr. Then the mixture was cooled to rt and diluted with CH$_2$Cl$_2$, washed with H$_2$O, 5% HCl, H$_2$O, and saturated aqueous NaHCO$_3$. The aqueous phase was extracted with ethyl acetate (15 mL ×3). The combined organic extracts were dried over Na$_2$SO$_4$, filtered and evaporated under reduced pressure. Silica gel flash column chromatography (ethyl acetate/hexanes = 3:1) gave a pale-yellow solid (47 mg, 37%) as the product. **PC9**: mp = 132–133℃. $^1$H NMR (400 MHz, CDCl$_3$) δ 8.58 (d, $J$ = 5.9 Hz, 2H), 7.44 (d, $J$ = 8.7 Hz, 2H), 7.36–7.29 (m, 2H), 7.17 (dd, $J$ = 15.5, 10.2 Hz, 1H), 6.99–6.74 (m, 4H), 6.57 (d, $J$ = 15.5 Hz, 1H), 4.11 (d, $J$ = 7.0 Hz, 2H), 1.48 (t, $J$ = 7.0 Hz, 3H). $^{13}$C NMR (101 MHz, CDCl$_3$) δ 159.0, 149.8, 144.7, 135.4, 133.8, 129.2, 128.2, 127.9, 125.8, 120.3, 114.6, 63.3, 14.6.HRMS (+ESI) m/z calcd. for C$_{17}$H$_{18}$NO (M + H)$^+$ 252.1383, found 252.1384.

## Preparation and quantification of the enzymes

A truncated form of SARM1, SARM1-dN, was prepared as described (*Zhao et al., 2019*). In brief, the recombinant protein, SARM1 without the N-terminal mitochondrial signal, was expressed in HEK293T cells and released by 100 µM digitonin in PBS with protease inhibitor cocktail (Roche). The cell lysate of wildtype HEK293T, prepared with the same method, was used as the negative control. To quantify SARM1-dN, the protein was pulled down by BC2 nanobody (*Bruce and McNaughton, 2017*) conjugated beads which were prepared by conjugating BC2 nanobody to NHS-beads (GE Healthcare). The purified SARM1-dN, named as SARM1-IP, together with the certain amounts of standard protein BSA, was applied to SDS-PAGE, which was stained by Coomassie blue. The protein contents of SARM1-dN were then calculated by the band intensity with BSA as standards.

DtSARM1-dN, with the N-terminal targeting signal removed and tagged with a tandem strep tag II and flag tag for purification, was constructed into Plenti-CMV-puro-Dest (Invitrogene) by LR clonase II enzyme according to the manufacturer's instructions. HEK293F cells overexpressing dtSARM1-dN were constructed by lentivirus infection and selected with 1 µg/mL puromycin. DtSARM1-dN was released by 200 µM digitonin and immunoprecipitated with StrepTactin resin (GE healthcare), washed with buffer W (100 mM Tris-HCl pH8.0, 150 mM NaCl and 1 mM EDTA) for four times and eluted with 2 mM biotin in buffer W. DtSARM1-dN was used in the experiments on the dHNN-modification, cysteine-to-alanine mutants and Cryo-EM structures.

Recombinant CD38 and *N. crassa* NADase were prepared as described previously (*Graeff et al., 1994*; *Munshi et al., 1997*).

## In vitro fluorescence assays

To analyze the activity of SARM1 with PCs in vitro, reactions were started by incubating the enzyme with the reaction mixture, 50 µM PC, 100 µM NAD, and 100 µM NMN in PBS. The absorbance and fluorescence were measured in a quartz cuvette or black 96-well plates (Corning), respectively, in an Infinite M200 PRO microplate reader (Tecan). For the assays with εNAD,NHD, or NGD as the substrate, 100 µM of each probe was incubated with the enzymes and the kinetics of fluorescence production was measured at $\lambda_{ex}$ = 300 nm, $\lambda_{em}$ = 410 nm. The initial rate of the reactions was quantified with the slope of the fluorescence increase in the first several minutes.

## HPLC analysis of the base-exchange reaction

The reactions were prepared by mixing SARM1-IP (SARM1 binding on BC2-beads, around 2.5 µg/mL) with 100 µM NAD, 50 µM PC6, 100 µM NMN, and 0.1 mg/mL BSA in PBS and incubated for 60 min at 37°C. SARM1-IP was removed by centrifugation at 4,500 rpm for 1 min. The cleaned mixtures were applied to a C-18 reverse phase column equipped on an HPLC (Agilent 1260) with a gradient of 0.1 M $KH_2PO_4$ (pH 6.0) and 0.1 M $KH_2PO_4$ (pH 6.0) with MeOH (7:3) to elute NMN, cAPPR, ADPR, NAD, and a gradient of ACN from 30 to 70% to elute PAD6 and PC6. The PAD6 fractions were collected and lyophilized for the characterization of absorption and fluorescence spectra.

To analyze PAD6 in cells, the metabolites were extracted from the cells treated with 50 µM PC6 by 0.6 M perchloric acid, followed by the neutralization with Chloroform: Tri-n-octylamine (3:1). The extracts were applied to a C-18 column and PAD6 was eluted with water and acetonitrile by 2% acetonitrile for 8 min, then 30% acetonitrile for 8 min.

## Confocal imaging of PAD6 in living cells

HEK293 cells, overexpressing wildtype or the enzymatically dead form (E642A) of SARM1 or HEK293T Knocking out *NMNAT1* were constructed as before (*Zhao et al., 2019*). Cells, grown on 0.05 mg/mL poly-L-lysine coated Chambered coverglass (ThermoFisher, #155411) overnight, were treated with 50 µM PC6 in the presence or absence of 100 µM CZ-48 for 8 hr (for SARM1-OE cells) and 200 µM CZ-48 for 48 hr (for wildtype HEK293T cells), respectively. To demonstrate the edges of the cells, they were stained with 50 µg/mL Concanavalin A, Alexa Fluor 647 Conjugate (ThermoFisher) at 4°C for 10 min before imaging. The fluorescence signals (Ex/Em: 405/525 nm for PAD6; Ex/Em: 561/590 for ConA) were captured under a confocal microscope (Nikon A1).

## Analysis of PAD6 signals in live cells by flow cytometry

HEK293 cell line carrying doxycycline (Dox)-inducible SARM1 was constructed as previously described (*Zhao et al., 2019*). The cells were treated with 50 µM PC6, 100 µM CZ-48, or 0.5 mg/mL Dox for 4, 8, 12, and 16 hr. The cells were trypsinized and the fluorescence of PAD6 (Ex/Em: 405/525 nm) was analyzed by flow cytometry (CytoFlex, Beckman).

## DRG culture and imaging

Mouse DRG culture was performed as described (*Sasaki et al., 2016*). Briefly, DRGs were dissected from the embryos at day 12.5 to 14.5 (E12.5-E14.5), dispersed by 0.05% Trypsin solution containing 0.02% EDTA (Gibco), and seeded in Neurobasal Plus Medium supplemented with 2% B27 plus, 1 mM GlutaMax, 1% penicillin/streptavidin solution, and 37.5 ng/mL NGF on the Chambered coverglass pre-coated with (0.1 mg/mL) poly-L-Lysine, (0.02 mg/mL) laminin, and 5% FBS. Every 3 days, 50% of the culture media was replaced by fresh media with the addition of 5 µM 5-fluoro-2'-deoxyuridine and 5 µM uridine.

On div6, the neurons were infected with lentivirus carrying various expression cassettes of genes or shRNAs. Three days later, the cells were treated with 50 µM PC6 in the absence or presence of 200 µM CZ-48 or 50 nM vincristine. The fluorescence images (Ex/Em: 405/520 nm for PAD6; Ex/Em: 561/590 for TdTomato) were captured under a confocal microscope (Nikon A1) with a 60x object. The mean fluorescence intensity was quantified by NIS-Elements AR analysis (Nikon). Axon degeneration was quantified based on axon morphology using ImageJ. The TdTomato fluorescence images were binarized and measured the total axon area (size = 20 infinity pixels) and the degenerated axon (size = 20–4,000,000 pixels) with particle analyzer module of ImageJ. Axon degeneration index was calculated as the ratio of the degeneration axon over total axon area.

Lentivirus preparation and infection of DRG neurons pLKO.1-shRNA-*Sarm1* plasmids were constructed as described previously (*Zhao et al., 2019*). Briefly, the shRNA targeting mouse *Sarm1* (5'-CCGGCTGGTTTCTTACTCTACGAATCTCGAGATTCGTAGAGTAAGAAACCAGTTTTTG-3') or the scrambled shRNA (5'- CCGGCCTAAGGTTAAGTCGCCCTCGCTCGAGCGAGGGCGACTTAACCTTAGGTTTTTG-3') were inserted to pLKO.1-puro (Addgene, #8453) with EcoRI and AgeI, followed by the replacement of the puromycin resistance gene with a fluorescent protein, TdTomato (GenBank: LC311026.1) with KpnI/BamHI. The lentiviral particles were prepared by transfecting HEK293T cells with the corresponding lentivectors, pMD2.G, and psPAX2 (*Liu et al., 2017*) and concentrated with Lenti-Concentin Virus Precipitation Solution (ExCell Bio). The viral particles were finally

resuspended in Neurobasal Plus Medium. The virus titer was determined by series infection of HEK293T cells. The virus was added to infect the DRG neurons on div6 with the same MOIs and the experiments were carried out 72 hr after infection.

### Imaging and quantification of AxD after axotomy and vincristine treatment

For axotomy, one DRG was seeded into a 24-well plate, and 5 µM 5-fluoro-2′-deoxyuridine and 5 µM uridine were added on the other day. On div5, axons were pre-incubated with the drugs for 0.5 hr and severed near the soma with a 3 mm flat blade under microscope guidance to remove the cell bodies. For vincristine treatment, DRGs were digested with 0.05% Trypsin and seeded into 24-well plates. DRGs on div9-13 were incubated with 50 nM vincristine in the presence or absence of the candidate drugs.

About 9–12 images of the axon were acquired in the bright field with a 20x object for each treatment at the indicated time points using invert optical microscope (Olympus). Axon degeneration was quantified using ImageJ. For each treatment, 60 random grid-squares with $147 \times 147$ pixels were cropped, binarized and the total axon area (size = 16 infinity pixels) and the degenerated axon (size = 16–10,000 pixels) were quantified with the particle analyzer module of ImageJ. Axon degeneration index was calculated as the ratio of the degeneration axon over total axon area.

### Measurement of the cADPR levels in DRGs

DRG neurons were treated with 50 nM Vincristine or 200 µM CZ-48 for 0, 12, 24, 48 hr on div6. After incubation, DRG was washed with cold PBS and lysed with 0.6 M perchloric acid. The concentration of cADPR was analyzed by the cycling assay, as described previously (*Graeff and Lee, 2002*).

### Q-RT-PCR

The total RNAs were extracted from DRG neurons with RNA extraction kit (OMEGA) 48 hr post infection and transcribed with the kit, Transcript II One-step gDNA Removal and cDNA synthesis Supermix from Sangon Biotech. The mRNA level of SARM1 relative to GAPDH was quantified with by Q-RT-PCR using TransStart Tip Green qPCR SuperMix (TransGen Biotech) on CFX Connect Real-Time PCR Detection System (Bio-Rad). The following primer pairs were used: *Sarm1* sense, 5′-C TTTCTCCAAGGAGGACGAGC-3′, antisense, 5′-CTTGTGTCACTGGCATCCACC-3′; GAPDH sense, 5′- TGGCCTTCCGTGTTCCTAC-3′, antisense, 5′-GAGTTGCTGTTGAAGTCGCA-3′.

### PC6 assay

For high-throughput screening of the potential inhibitors, 1.5 µg/mL SARM1-dN was pre-incubated with 50 µM drugs (TargetMol, L1000) at room temperature for 10 min and the reaction reagents, including 20 µM PC6, 50 µM NAD, and 50 µM NMN were added to start the reaction. Controls including reactions without the drugs, defined as 0% inhibition, and without both the drugs and SARM1-dN, defined as 100% inhibition. The fluorescence (ex: 390 nm; em: 520 nm) was measured by plate reader (Tecan, M200Pro) and the initial reaction rates were calculated, $V_x$ for the reactions with different drugs, $V_{max}$ for the reaction without drugs and $V_{min}$ for the reaction without enzyme. The inhibitory rates were calculated by the equation, $(V_{max} - V_x)/V_{max}$ and plotted using GraphPad Prism 8.0. The standard statistics of the screening were calculated as follows: Z′ factor=1-(3*SD($V_{max}$)+3*SD($V_{min}$))/(Average($V_{max}$)-Average($V_{min}$)), S/N = (Average($V_{max}$)-Average($V_{min}$))/SD($V_{min}$). In the screening of this study, Z′ = 0.69 and S/N = 291.96.

For $IC_{50}$ measurement, 0.4 µg/mL SARM1-dN was pre-incubated with doses of drugs in vitro for 10 min, and started the reaction by adding 50 µM NAD, 50 µM NMN, and 50 µM PC6. Calculation of $IC_{50}$ by plotting the initial rate to dose of compounds.

### NADase acitivity of SARM1 analyzed by HPLC

1 µg/mL of SARM1-dN was pre-incubated with drugs for 15 min at room temperature, and the reactions were started by adding 100 µM NAD and 100 µM NMN. They were stopped by removing the enzyme with MultiScreen Filter Plates (Millipore) after 0, 15, and 30 min incubation at 37°C and the reactants and products were analyzed by a C-18 column (Aligent, 20 RBAX SB-C18) with a gradient of 0.1 M $KH_2PO_4$ (pH 6.0) and 0.1 M $KH_2PO_4$ (pH 6.0) with MeOH (7:3) to elute NMN, cAPPR,

ADPR, NAD, Nam. The amount of ADPR was used to calculate the initial rate. $IC_{50}$ was calculated by Graphpad Prism 8.0.

## HPLC analysis of NSDP and its derivatives

The NSDP powder was dissolved in DMSO and shined with UV at 254 nm for 30 min and analyzed with a C-18 reverse phase column (ZORBAX SB-C18) equipped on a HLPC (Aligent 1260) and eluted with 50% of 0.1%TFA and 50% of 0.1%TFA in 99% ACN. The products after UV treatment were collected and purified by HPLC, as described above. The inhibitory activity of these fractions was determined by PC6 assay after being neutralized with 100 mM Tris (pH7.5), and the main peak was characterized by HRMS (Thermo, Q Exactive Focus).

## The inhibitory activity of dHNN in vitro and in cellulo

To determine whether dHNN inhibits the activation or enzymatic activity of SARM1 in vitro, SARM1-dN, the autoinhibited form, and SAM-TIR, the constitutively active form, were pre-incubated with different concentrations of dHNN at rt for 10 min, after which the activity was measured with PC6 assay and the inhibition rate was calculated.

To test the same effect in cellulo, HEK293 cells overexpressing the inducible SARM1 (iSARM1) or SAM-TIR (iSAM-TIR) were pre-incubated with 20 µM dHNN, or DMSO as controls, for 1.5 hr and then treated with 100 µM CZ-48 or 0.5 µg/mL Dox for the indicated time. The cellular levels of cADPR were measured as described above.

## Modification of SARM1 by dHNN

The dtSARM-dN, eluted from the StrepTactin beads, was incubated with 0, 5, or 50 µM dHNN at rt for 40 min and applied to SDS-PAGE. After simplyBlue SafeStain (ThermoFisher), the dtSARM1-dN band was sliced, dehydrated with 100% ACN, and the proteins were alkylated by 22.5 mM IAA for 30 min in dark after the reduction by 10 mM DTT at 55°C for 30 min. After overnight in-gel digestion with Trypsin, the peptides were extracted and analyzed with HRMS (Thermo, Q Exactive HF-X). The dHNN modifications, determined by Protein Discoverer software (ThermoFisher), were defined as an increase of molecular weight by 370.153 Da, 354.158 Da, 402.143 Da, or 386.148 Da on the cysteine residues characterized in the MS[2] fragments (*Callan et al., 2009*; *Möller et al., 2017*). The abundance of each peptide was determined in the MS[1]. Abundance ratio was calculated by dividing the intensity of the dHNN-modified peptides by that of the corresponding peptides.

## Cysteine mutants

The cysteine-to-alanine mutants of dtSARM1-dN were amplified by the overlapping PCRs with the primers, listed below, and subcloned into pCDH-EF1-MCS-IRES-neo by Xba I and Not I. To prepare the mutant proteins, HEK293 cells were transfected with the above plasmids by lipofectamine 2000 or Polyethylenimine according to the manufacturer's instructions, and the proteins were extracted 48–72 hr after transfection and determined the $IC_{50}$ of dHNN by PC6 assay in vitro.

```
C117A-F: 5'-GTAGCCCAGGGTCTGGCC GACGCCATCCGC-3'
C117A-R: 5'-GCGGATGGCGTCGGCCAGACCCTGGGCTAC-3'
C199A-F: 5'-CATTCGGAGGAGACAGCC CAGAGGCTGGTG-3'
C199A-R: 5'-CACCAGCCTCTGGGCTGTCTCCTCCGAATG-3'
C215A-F: 5'-GCGGTGCTGTATTGGGCACGCCGCACGGAC-3'
C215A-R: 5'-GTCCGTGCGGCGTGCCCAATACAGCACCGC-3'
C226A-F: 5'-GCGCTGCTGCGCCACGCAGCGCTGGCGCTG-3'
C226A-R: 5'-CAGCGCCAGCGCTGCGTGGCGCAGCAGCGC-3'
C233A-F: 5'-CTGGCGCTGGGCAACGCAGCGCTGCACGGG-3'
C233A-R: 5'-CCCGTGCAGCGCTGCGTTGCCCAGCGCCAG-3'
C271A-F: 5'-CTTCGGCTGCACGCCGCACTCGCAGTAGCG-3'
C271A-R: 5'-CGCTACTGCGAGTGCGGCGTGCAGCCGAAG-3'
C311A-F: 5'-GGCCGCTTCGCCCGCGCC CTGGTGGACGCC-3'
C311A-R: 5'-GGCGTCCACCAGGGCGCGGGCGAAGCGGCC-3'
C343A-F: 5'-CGCTTGGAGGCGCAGGCAATCGGGGCTTTC-3'
C343A-R: 5'-GAAAGCCCCGATTGCCTGCGCCTCCAAGCG-3'
C350A-F: 5'-GGGGCTTTCTACCTCGCAGCCGAGGCTGCC-3'
```

C350A-R: 5'-GGCAGCCTCGGCTGCGAGGTAGAAAGCCCC-3'
C430A-F: 5'-GGTTTCTCCAAGTACGCAGAGAGCTTCCGG-3'
C430A-R: 5'-CCGGAAGCTCTCTGCGTACTTGGAGAAACC-3'
C482A-F: 5'-GCCAACTATTCTACGGCC GACCGCAGCAAC-3'
C482A-R: 5'-GTTGCTGCGGTCGGCCGTAGAATAGTTGGC-3'
C508A-F: 5'-TACGGCCTGGTCAGCGCAGGCCTGGACCGC-3'
C508A-R: 5'-GCGGTCCAGGCCTGCGCTGACCAGGCCGTA-3'
C527A-F: 5'-CAGCTGCTGGAAGACGCAGGCATCCACCTG-3'
C527A-R: 5'-CAGGTGGATGCCTGCGTCTTCCAGCAGCTG-3'
C552A-F: 5'-CACTCCCCGCTGCCCGCAACTGGTGGCAAAC-3'
C552A-R: 5'-GTTTGCCACCAGTTGCGGGCAGCGGGGAGTG-3'
C629A-F: 5'-GGAGCACTGGACAAGGCAATGCAAGACCAT-3'
C629A-R: 5'-ATGGTCTTGCATTGCCTTGTCCAGTGCTCC-3'
C635A-F: 5'-ATGCAAGACCATGACGCAAAGGATTGGGTG-3'
C635A-R: 5'-CACCCAATCCTTTGCGTCATGGTCTTGCAT-3'
C649A-F: 5'-GTGACTGCTTTAAGCGCC GGCAAGAACATT-3'
C649A-R: 5'-AATGTTCTTGCCGGCGCTTAAAGCAGTCAC-3'
dtSARM1-dN-F: 5'-CAGTCTAGAATGGACTACAAGGATGACGATG-3'
dtSARM1-dN-R: 5'-ATAGCGGCCGCTTAGGTTGGACCCA-3'

## Western blots

Cells were lysed with RIPA buffer (50 mM Tris-HCl, 150 mM NaCl, 1 mM EDTA and 0.05% Triton, pH 7.4). Each sample was loaded onto 10–12% SDS-PAGE gels and the proteins were then transferred to a PVDF membranes. The membranes were blocked with 5% milk and blotted with anti-SARM1 (home-made), with anti-Tubulin (TransGen Biotech) as an internal control. After incubation with HRP-conjugated second antibodies, the signals were developed by ECL (Abvansta), detected, and quantified by a Chemidoc MP system and ImageLab software (Bio-Rad).

## Cryo-EM sample preparation, data collection, and processing

Pure dtSARM1-dN protein was concentrated to 3 mg/mL and pre-incubated with 50 µM dHNN at rt for 10 min, and applied to glow-discharged gold grid, blotted in FEI Vitrobot Mark IV (ThermoFisher Scientific) before frozen by liquid ethane and stored in liquid nitrogen. The sample without inhibitor was examined at the Cryo-EM center of Chinese University Hong Kong (Shenzhen) on a 300kV Titan Krios (ThermoFisher Scientific) equipped with Gatan K3 direct electron detector under magnification of 105,000x, with the corresponding pixel size of 0.85 Å. The dose rate was set to 17.6 e/pix/s and exposure time was set to 2.5 s to obtain 50 frames, which led to an accumulated dose of 61 electrons per $Å^2$. The total dataset consists of 2692 raw movies with a defocus value range of −1.0 to −2.0 µm. Motion correction and CTF parameter estimation were performed with cryoSPARC (*Punjani et al., 2017*). 2,012,198 particles were autopicked. After several rounds of 2D classification, 712,139 particles were selected for generation of the final 2D average results.

The dHNN-treated sample was examined at the Cryo-EM center of Southern University of Science and Technology on a Titan Krios G3 (ThermoFisher Scientific) with Gatan K2 summit detector with a nominal magnification of 130,000x and corresponding pixel size of 1.076 Å. A total accumulative dose of 50 e⁻/$Å^2$ was set for each exposure and split into 39 frames during data acquisition. The defocus range was set between −0.8 and −2.0 µm. In total, 2890 images were collected. Motion correction and CTF parameter estimation were performed with MotionCor2 and CTFFind4 built within Relion 3.1 (*Fernandez-Leiro and Scheres, 2017*). After CTF estimation, images with thick ice, obvious shift or cleft were removed, which left 2673 images for further processing. 2,655,835 particles were autopicked from these images. After several rounds of 2D classification, 700,472 particles were selected and exported for generation of the final 2D average results with CryoSparc and 3D refinement with CisTEM beta-1.0.0 (*Grant et al., 2018*). The particle stack was subject to 10 rounds of 3D auto-refinement among 6 classes using 6WPK as initial model. Four classes with higher estimated resolution were selected and combined for 20 more rounds of 3D manual global refinement and one class with the highest occupation (62.5%) and best resolution was chosen for several more rounds of 2D and 3D classification with Relion 3.1 and CisTEM beta-1.0.0. The resolution for the final map was around 2.4 Å.

The previously reported structures of the SARM1 SAM domain (PDB: 6O0S) and TIR domain (PDB: 6O0Q) were used as model templates during initial model building. The initial model of ARM domain was built de novo in Coot (*Emsley et al., 2010*). The three domains of SARM1 were connected in Coot and docked into density maps using Dock in Map module of Phenix 1.16 (*Adams et al., 2010*) with C8 symmetry and then subjected to multiple rounds of Real-space refinement in Phenix. The dHNN molecule was built and fitted into the density around Cys311 of the initial model in Coot. The final models were validated with Comprehensive Validation module of Phenix and the refinement statistics are listed in *Supplementary file 1*. The model and EM map have been deposited in Protein Data Bank with accession codes of PDB ID 7DJT and EMD-30700.

## Data analysis

All experiments contained at least three biological replicates. Data shown in each figure are all means ± SD. The unpaired Student's *t*-test was used to determine statistical significance of differences between means (*$p < 0.05$, **$p < 0.01$, ***$p < 0.001$, ****$p < 0.0001$). GraphPad Prism 8.0 was used for data analysis.

# Acknowledgements

We would like to thank the Cryo-EM center of Southern University of Science and Technology for Cryo-EM data collection and the HPC-Service Station in Cryo-EM center of Southern University of Science and Technology for data processing. We acknowledge Beijing Artivila Biopharma Co. Ltd for the supports.

# Additional information

## Competing interests

Yong Juan Zhao: Two Chinese patents (202010528147.3; 202011359354.7) are in the process of application. The other authors declare that no competing interests exist.

## Funding

| Funder | Grant reference number | Author |
| --- | --- | --- |
| Ministry of Science and Technology of the People's Republic of China | 2019YFA0906000 | Hongmin Zhang |
| National Science Foundation of China | 31871401 | Yong Juan Zhao |
| Hong Kong Baptist University | RC-SGT2/18-19/SCI/005 | Chi-Sing Lee |
| Hong Kong Baptist University | RC-ICRS-18-19-01A | Chi-Sing Lee |

The funders had no role in study design, data collection and interpretation, or the decision to submit the work for publication.

## Author contributions

Wan Hua Li, Conceptualization, Data curation, Formal analysis, Validation, Investigation, Visualization, Methodology, Writing - original draft, Writing - review and editing; Ke Huang, Data curation, Validation, Investigation; Yang Cai, Data curation, Formal analysis, Validation, Visualization, Methodology; Qian Wen Wang, Data curation, Formal analysis, Validation; Wen Jie Zhu, Yun Nan Hou, Suj-ing Wang, Zhi Ying Zhao, Xu Jie Xie, Data curation, Formal analysis, Validation, Methodology; Sheng Cao, Data curation, Formal analysis, Validation, Visualization; Yang Du, Formal analysis, Methodology, Writing - review and editing; Chi-Sing Lee, Conceptualization, Resources, Data curation, Formal analysis, Supervision, Funding acquisition, Validation, Investigation, Methodology, Writing - original draft, Writing - review and editing; Hon Cheung Lee, Conceptualization, Resources, Data curation, Formal analysis, Supervision, Validation, Investigation, Visualization, Methodology, Writing - original draft, Writing - review and editing; Hongmin Zhang, Resources, Data curation, Formal analysis,

Funding acquisition, Validation, Investigation, Visualization, Methodology, Writing - review and editing; Yong Juan Zhao, Conceptualization, Resources, Data curation, Formal analysis, Supervision, Funding acquisition, Validation, Investigation, Visualization, Methodology, Writing - original draft, Project administration, Writing - review and editing

#### Author ORCIDs

Wan Hua Li [iD] https://orcid.org/0000-0002-3881-5796
Chi-Sing Lee [iD] https://orcid.org/0000-0002-3564-8224
Hon Cheung Lee [iD] https://orcid.org/0000-0002-6993-0121
Hongmin Zhang [iD] https://orcid.org/0000-0003-4356-3615
Yong Juan Zhao [iD] https://orcid.org/0000-0003-4564-1912

#### Ethics

Animal experimentation: This study was carried out in strict accordance with animal use protocol approved by Peking University Shenzhen Graduate School Animal Care and Use Committee (#AP0015001). All animals (C57BL6/J), purchased from Guangdong Medical Laboratory Animal Center (China), were handled in accordance with the guidelines of the Committee on the Ethic of Animal Experiments. All surgery was performed after euthanasia and efforts were made to minimize suffering.

#### Decision letter and Author response

Decision letter https://doi.org/10.7554/eLife.67381.sa1
Author response https://doi.org/10.7554/eLife.67381.sa2

## Additional files

#### Supplementary files

• Supplementary file 1. Refinement statistics for the SAMR1-dHNN structure.

• Transparent reporting form

#### Data availability

Diffraction data have been deposited in PDB under the accession code 7DJT. All data generated or analysed during this study are included in the manuscript and supporting files.

The following dataset was generated:

| Author(s) | Year | Dataset title | Dataset URL | Database and Identifier |
|---|---|---|---|---|
| Cai Y, Zhang H | 2021 | Human SARM1 inhibitory state bounded with inhibitor dHNN | https://www.rcsb.org/structure/7DJT | RCSB Protein Data Bank, 7DJT |

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
