## [Decision Letter]

**Acceptance summary:**

This is a very nice manuscript describing the development of small molecule probes that can detect the activity of SARM1 in live cells. One of the probe allows the author to monitor the activity SARM1 and discovery a small molecule derived from an hyertension drug as an effective SARM1 inhibitor. SARM1 is an enzyme that can convert NAD+ to ADP-ribose or cyclic ADP-ribose and is implicated in axon degeneration, thus the probe and the inhibitor described in the manuscript could lead to future therapeutic development targeting SARM1 to treat axon degeneration.

**Decision letter after peer review:**

Thank you for submitting your article "Permeant Fluorescent Probes Visualize the Activation of SARM1 and Uncover an Anti-neurodegenerative Drug Candidate" for consideration by *eLife*. Your article has been reviewed by 2 peer reviewers, including Hening Lin as the Reviewing Editor and Reviewer #1, and the evaluation has been overseen by Michael Marletta as the Senior Editor.

Essential revisions:

1. Since the inhibitors react with multiple cysteines and the mechanism of inactivation is not clear from the MS data, please either propose a reasonable mechanism and/or minimize those aspects of the report and explicitly state that the inhibitor modifies multiple cysteines and is unlikely to be a very useful probe.

2. It would be necessary to do a more thorough editing and proof-reading to further improve the language.

*Reviewer #1:*

The authors aimed to develop cell-permeable small molecule probes that can monitor the activity of SARM1, an enzyme that hydrolyzes NAD+ and is thought to be important for axon degeneration. They successfully achieved this goal using the base exchange activity of SARM1 to make a donor-π-acceptor type of fluorophore. The best probe described in the manuscript is PC6. A number of experiments were carried to rigorously test that the probe works as expected. PC6 has a number of nice features. It is cell permeable, gives much stronger signal than any other probes known for SARM1, is specific for SARM1 and does not detect the activity of CD38 (another enzyme that has similar activity), and allows detection of endogenous SARM1 activation in neurons.

Using this probe PC6, the authors was able to monitor SARM1 activity in neurons treated with vincristine and demonstrated that SARM1 activation precedes axon degeneration and is important but not sufficient for axon degeneration. Most importantly, using this probe to monitor SARM activity, they screened a library of about 2000 drug molecules and discovered that a hypertension drug, nisoldipine, could inhibits SARM1. Surprisingly, further studies showed that a derivative of nisoldipine, dehydronitrosonisoldipine (dHNN, present in the nisoldipine compound used ), is actually the inhibitor of SARM1. They then carried nice mechanistic studies (including mass spectrometry and cryo-EM structures) showing that dHNN inhibits SARM1 by covalently modify Cys311 residue in the ARM domain. The dHNN binding site is similar to the previously established NAD+ inhibitory site.

Overall, the probe is novel with many useful features, the study is rigorous and rather complete, and the conclusion is well supported. I believe the study will be important for the field and will be well received by the field.

*Reviewer #2:*

The manuscript by Li et al. describes the development of styrylpyridines as cell permeant fluorescent sensors of SARM1 activity. This work is significant because SARM1 activity is increased during neuron damage and SARM1 knockout mice are protected from neuronal degeneration caused by a variety of physical and chemical insults. Thus, SARM1 is a key player in neuronal degeneration and a novel therapeutic target. SARM1 is an NAD+ hydrolase that cleaves NAD+ to form nicotinamide and ADP ribose (and to a small extent cyclic ADP ribose) via a reactive oxocarbenium intermediate. Notably, this intermediate can either react with water (hydrolysis), the adenosine ring (cyclization to cADPR), or with a pyridine containing molecule in a 'base-exchange reaction'. The styrylpyridines described by Li et al. exploit this base-exchange reaction; the styrylpyridines react with the intermediate to form a fluorescent product. Notably, the best probe (PC6) can be used to monitor SARM1 activity in vitro and in cells. Upon validating the utility of PC6, the authors use this compound to perform a high throughput screen of the Approved Drug Library (L1000) from TargetMol and identify nisoldipine as a hit. Further studies revealed that a minor metabolite, dehydronitrosonisoldipine (dHNN), is the true inhibitor, acting with single digit μM potency. The authors provide structural and proteomic data suggesting that dHNN inhibits SARM1 activity via the covalent modification of C311 which stabilizes the enzyme in the autoinhibited state.

Key strengths of the manuscript include the probe design and the authors demonstration that they can be used to monitor SARM1 activity in vitro in an HTS format and in cells. The identification of C311 as potential reactive cysteine that could be targeted for drug development is an important and significant insight.

Key weaknesses include the fact that dHNN is a highly reactive molecule and the authors note that it modifies multiple sites on the protein (they mentioned 8 but MS2 spectra for only 5 are provided). As such, the compound appears to be a non-specific alkylator that will have limited utility as a SARM1 inhibitor. Additionally, no information is provided on the proteome-wide selectivity of the compound. An additional key weakness is the lack of any mechanistic insights into how the adducts are generated.

Moreover, it is not clear how the proposed sulphonamide and thiohydroxylamine adducts are formed. From the images presented, it is unclear whether there is sufficient 'density' in the cryoEM maps to accurately predict the sites of modification. Finally, the authors do not show whether the conversion of PC6 to PAD6 is stable or if PAD6 can also be hydrolyzed to form ADPR.

---

## [Author Response]

Essential revisions:1. Since the inhibitors react with multiple cysteines and the mechanism of inactivation is not clear from the MS data, please either propose a reasonable mechanism and/or minimize those aspects of the report and explicitly state that the inhibitor modifies multiple cysteines and is unlikely to be a very useful probe.

Although dHNN may react with multiple cysteines, our results indicate the primary one is Cys311. Nevertheless, we have added the qualification that other Cys may contribute to the inhibition (c.f. Abstract, the second paragraph on p.17). The evidence for Cys311 being dominant includes quantification of the intensity of the modified peptides and normalizing with that of the corresponding total peptides, with or without modification, showing that the modification is mainly on Cys311 (Figure 5—figure supplement 1). The dominant role of Cys311 is also confirmed by our mutagenesis and structural studies. Furthermore, quantification of cysteine-containing peptides of other proteins showed no dHNN modification. So, we conclude that dHNN shows significant specificity to the Cys311 of SARM1. The inhibitory mechanism of dHHN we propose is like that of NAD, prohibiting the conformational change of the ARM domain by acting on the allosteric site that includes Cys311 to inhibit the activation of SARM1.

2. It would be necessary to do a more thorough editing and proof-reading to further improve the language.

We have carefully edited and improved our manuscript throughout.

Reviewer #2:The manuscript by Li et al. describes the development of styrylpyridines as cell permeant fluorescent sensors of SARM1 activity. This work is significant because SARM1 activity is increased during neuron damage and SARM1 knockout mice are protected from neuronal degeneration caused by a variety of physical and chemical insults. Thus, SARM1 is a key player in neuronal degeneration and a novel therapeutic target. SARM1 is an NAD+ hydrolase that cleaves NAD+ to form nicotinamide and ADP ribose (and to a small extent cyclic ADP ribose) via a reactive oxocarbenium intermediate. Notably, this intermediate can either react with water (hydrolysis), the adenosine ring (cyclization to cADPR), or with a pyridine containing molecule in a 'base-exchange reaction'. The styrylpyridines described by Li et al. exploit this base-exchange reaction; the styrylpyridines react with the intermediate to form a fluorescent product. Notably, the best probe (PC6) can be used to monitor SARM1 activity in vitro and in cells. Upon validating the utility of PC6, the authors use this compound to perform a high throughput screen of the Approved Drug Library (L1000) from TargetMol and identify nisoldipine as a hit. Further studies revealed that a minor metabolite, dehydronitrosonisoldipine (dHNN), is the true inhibitor, acting with single digit μM potency. The authors provide structural and proteomic data suggesting that dHNN inhibits SARM1 activity via the covalent modification of C311 which stabilizes the enzyme in the autoinhibited state.Key strengths of the manuscript include the probe design and the authors demonstration that they can be used to monitor SARM1 activity in vitro in an HTS format and in cells. The identification of C311 as potential reactive cysteine that could be targeted for drug development is an important and significant insight.Key weaknesses include the fact that dHNN is a highly reactive molecule and the authors note that it modifies multiple sites on the protein (they mentioned 8 but MS2 spectra for only 5 are provided). As such, the compound appears to be a non-specific alkylator that will have limited utility as a SARM1 inhibitor. Additionally, no information is provided on the proteome-wide selectivity of the compound.

Although dHNN may react with cysteines in general, our results indicate it does target specifically Cys311. Quantification of cysteine-containing peptides of other proteins showed no dHNN modification. So, we conclude that dHNN shows significant specificity to the Cys311 of SARM1. Some other SH-reactive agents we tested show little inhibition on SARM1. The evidence for Cys311 being dominant includes quantification of the intensity of the modified peptides and normalizing with that of the corresponding total peptides, with or without modification, showing that the modification is mainly on Cys311 (Figure 5—figure supplement 1). The dominant role of Cys311 is also confirmed by our mutagenesis and structural studies. Our result strongly suggested that the C311 is a druggable site for designing allosteric inhibitors against SARM1 activation.

dHNN is effective in inhibiting SARM1 activation and AxD at low μM range, making it a useful inhibitor. Considering that the neuroprotective effect of NSDP, an approved drug, may well be due to dHNN, labeling it as inhibitor of SARM1 serves focus more attentions.

Revision has been made in Discussion.

An additional key weakness is the lack of any mechanistic insights into how the adducts are generated. Moreover, it is not clear how the proposed sulphonamide and thiohydroxylamine adducts are formed.

Please refer to Figure 1 in the paper (Sci Rep. 2017 Nov 1;7(1):14794.), which has been cited in our manuscript.

From the images presented, it is unclear whether there is sufficient 'density' in the cryoEM maps to accurately predict the sites of modification.

Please refer to Figure 5F, in which we show the close-up view of dHNN in the ARM domain. dHNN (purple) linked to the residue C311 and formed the hydrophobic interactions with surrounding residues E264, L268, R307, F308, and A315. The extra electron densities near the residue C311 fit the shape of dHNN and were shown as grey mesh.

Finally, the authors do not show whether the conversion of PC6 to PAD6 is stable or if PAD6 can also be hydrolyzed to form ADPR.

PAD6 is stable and cannot be hydrolyzed by the activated SARM1, as shown in Author response image 1. The reactions contain 10μM PAD6, 100 μM NMN, 2.65 μg/mL SARM1 or blank as a control. The PAD6 fluorescence was monitored for one hour and did not change in both groups.
